# GlomSpheres as a 3D co-culture spheroid model of the kidney glomerulus for rapid drug-screening

Jack Tuffin [1✉], Musleeha Chesor [1,2], Valeryia Kuzmuk[1], Tim Johnson[3], Simon C. Satchell[1],
Gavin I. Welsh [1] & Moin A. Saleem [1]

The glomerulus is the filtration unit of the kidney. Injury to any component of this specialised structure leads to impaired filtration and eventually fibrosis and chronic kidney disease. Current two and three dimensional (2D and 3D) models that attempt to recreate structure and interplay between glomerular cells are imperfect. Most 2D models are simplistic and unrepresentative, and 3D organoid approaches are currently difficult to reproduce at scale and do not fit well with current industrial drug-screening approaches. Here we report a rapidly generated and highly reproducible 3D co-culture spheroid model (GlomSpheres), better demonstrating the specialised physical and molecular structure of a glomerulus. Co-cultured using a magnetic spheroid formation approach, conditionally immortalised (CI) human podocytes and glomerular endothelial cells (GEnCs) deposited mature, organized isoforms of collagen IV and Laminin. We demonstrate a dramatic upregulation of key podocyte (podocin, nephrin and podocalyxin) and GEnC (pecam-1) markers. Electron microscopy revealed podocyte foot process interdigitation and endothelial vessel formation. Incubation with pro-fibrotic agents (TGF-β1, Adriamycin) induced extracellular matrix (ECM) dysregulation and podocyte loss, which were attenuated by the anti-fibrotic agent Nintedanib. Incubation with plasma from patients with kidney disease induced acute podocyte loss and ECM dysregulation relative to patient matched remission plasma, and Nintedanib reduced podocyte loss. Finally, we developed a rapid imaging approach to demonstrate the model's usefulness in higher throughput pharmaceutical screening. GlomSpheres therefore represent a robust, scalable, replacement for 2D in vitro glomerular disease models.

[1] Bristol Renal, Translational Health Sciences, Bristol Medical School, University of Bristol, Bristol BS1 3NY, UK. [2] Faculty of Medicine, Princess of Naradhiwas University, Narathiwat, Thailand. [3] UCB, Slough SL1 3WE, UK. ✉email: Jack.tuffin@bristol.ac.uk

Cell-based assays are an essential element of disease modelling and drug discovery. 2D cultures do not necessarily reflect the complex microenvironment cells encounter in a tissue, in particular the interconnections between different cell types and the extracellular matrix (ECM) surrounding them. Much of our knowledge comes from studies of the interaction of tumour cells with ECM, showing that adhesion and signalling is intimately affected by these interactions, leading for example to heterogeneous drug responses[1]. Within a tissue, molecular concentration gradients affect various cell behaviours, including motility, migration, and signalling, which cannot be replicated in 2D models[2].

Despite the progress made in advanced in vitro models, the early stages of drug development are still reliant on higher throughput 2D cell models. Utilising more advanced models only for later stages is insufficient, as false positives are compounded at each step of the drug discovery pipeline[3]. This highlights the importance of refining the earliest stages with better models, so that the right therapeutic candidates reach more advanced models.

3D models can be divided into scaffold-based or non-scaffold based technologies, the former being less adaptable for miniaturisation and/or high throughput screening. Non-scaffold technologies are rapidly advancing, relying on self-aggregation of cells in specialised environments[4]. To date these have mostly been used for tumour cells or have incorporated stem cells[5]. Co-culture models derived from mature cells, which represent a direct and rapid route towards a more differentiated physiological tissue structure, are lacking.

The glomerulus is a complex filter that allows the retention of cells and high molecular weight proteins in the blood, whilst allowing passage of small molecules, and electrolytes into the primary filtrate. The cell types making up the glomerular filtration barrier are glomerular endothelial cells and podocytes, on either side of a specialised basement membrane.

Conditionally immortalised cell lines of human podocytes[6], GEnCs[7] and mesangial cells[8] are the current gold standard in vitro tools and represented a step change in differentiation profile of cultured glomerular cells. These cells are however routinely mono-cultured in 2D environments, which (for podocytes at least) has resulted in some degree of phenotypic drift[9]. The formation of kidney organoids from induced-pluripotent stem cells is proving to be a promising step in the right direction for in vitro models[10,11]. It has since been shown that it is possible to derive glomeruli and primary podocytes from these organoids which show potential for toxicity screening[12]. These latter models are currently limited by lengthy protocols to generate the models, and phenotypic variability between experiments.

There is, therefore, an urgent need for better, consistent in vitro models of glomerular disease, to test specific injuries and their downstream consequences. A key disease example is of immune-mediated forms of nephrotic syndrome (NS), whereby a putative circulating factor directly damages the podocyte, leading to detachment, effacement and changes to the ECM, and to date has been modelled in vitro with human podocyte 2D cultures[13,14].

The intricacy of glomerular disease is difficult to model in vitro and much of our understanding thus far has been established by animal models and studies of human biopsy tissue. In vivo models of disease are limited by throughput, making them unsuitable for the early stages of drug development. For this reason, reductionist 2D cell models of disease are employed in pre-clinical testing, whereby they are used to narrow down libraries of >5000 compounds[15]. Many of these models are monoculture, and thus fail to replicate the intricate crosstalk environment of the glomerular filtration barrier that is understood to be important in glomerular disease progression[16,17].

Here we describe a scalable spheroid model of glomerular disease (GlomSpheres), capable of modelling 3D ECM dysregulation, podocyte loss and upscaling for automated compound screening. By using magnetic spheroid formation to co-culture conditionally immortalised human podocytes and GEnCs, cells deposit an organised 3D matrix of collagen IV, laminin and fibronectin between them. Moreover, GlomSpheres exhibit an ECM switch, whereby mature collagen IVα3 and laminin α5 are deposited. Biochemical analysis revealed that GlomSphere culture massively upregulates expression of characteristic markers for podocytes (nephrin, podocin and podocalyxin) and GEnC (pecam-1) when compared to 2D cultured counterparts.

Morphologically, scanning and transmission electron microscopy (SEM and TEM) demonstrate evidence of podocyte foot-process and basement membrane formation. Confocal imaging reveals a (PECAM-1 positive) endothelial vessel-like network, surrounded by podocin and nephrin-expressing podocytes. Incubation with fibrotic agents (TGF-β1/Adriamycin) is shown to induce ECM protein (collagen IV) dysregulation and podocyte loss, which are attenuated by the addition of a commercially available anti-fibrotic agent (Nintedanib). Incubation with NS relapse and remission patient plasma, modelling direct podocyte injury, demonstrates a loss of podocytes and increase of collagen IV deposition in relapse conditions relative to remission. Co-incubation with Nintedanib attenuates this podocyte loss but not collagen IV deposition, permitting dissection of disease mechanisms.

Finally, we developed a medium/high throughput method measuring podocyte integrity in a GlomSphere model of a human genetic mutation and demonstrate rescue by chemical chaperone compounds that can be analysed rapidly in a 96-well plate.

As such, GlomSpheres represent a robust, scalable replacement for 2D in vitro glomerular disease models with much improved disease homology.

## Results

**Formation and self-organisation of GlomSpheres**. GlomSpheres are generated by forming a core of 5000 GEnCs, which is then wrapped with a peripheral coating of 5000 podocytes (Supplementary Fig. 1a). Over a 10-day differentiation the wrapping of peripheral podocytes gradually migrates to more completely enclose the GEnC core (Supplementary Fig. 1b). Preliminary experiments, whereby spheroids were formed by completely mixing GEnCs and podocytes, showed that cells self-organise over 72 h (Supplementary Fig. 2). The reorganised structures have a GEnC core and peripheral podocytes. The decision was made to induce this organisation manually for reproducibility via layering spheroids (as in Supplementary Fig. 1) and differentiating for 10 days.

**Deposition of organised ECM proteins**. Immunofluorescent staining of paraffin embedded spheroid sections revealed a basement membrane-like layer of collagen IV, which is the primary constituent of the GBM (Fig. 1c). This staining is present extracellularly in both cell types but is primarily located at the interface between the two, as would be the GBM in vivo (Fig. 1a–d). When spheroids are wholemount stained, collagen IV organisation is more difficult to visualise, but podocyte organisation at the surface is visible (Fig. 1e). Staining of fibronectin reveals it to be most present at the podocyte/GEnC interface, although it is highly expressed extracellularly by both cell types (Supplementary Fig. 3).

**Switching to mature GBM proteins**. Wholemount IF staining (which compared mono-cultured podocytes and GEnCs to their co-cultured, GlomSphere counterparts) examined the presence of mature and immature GBM proteins (Fig. 2). It is apparent that

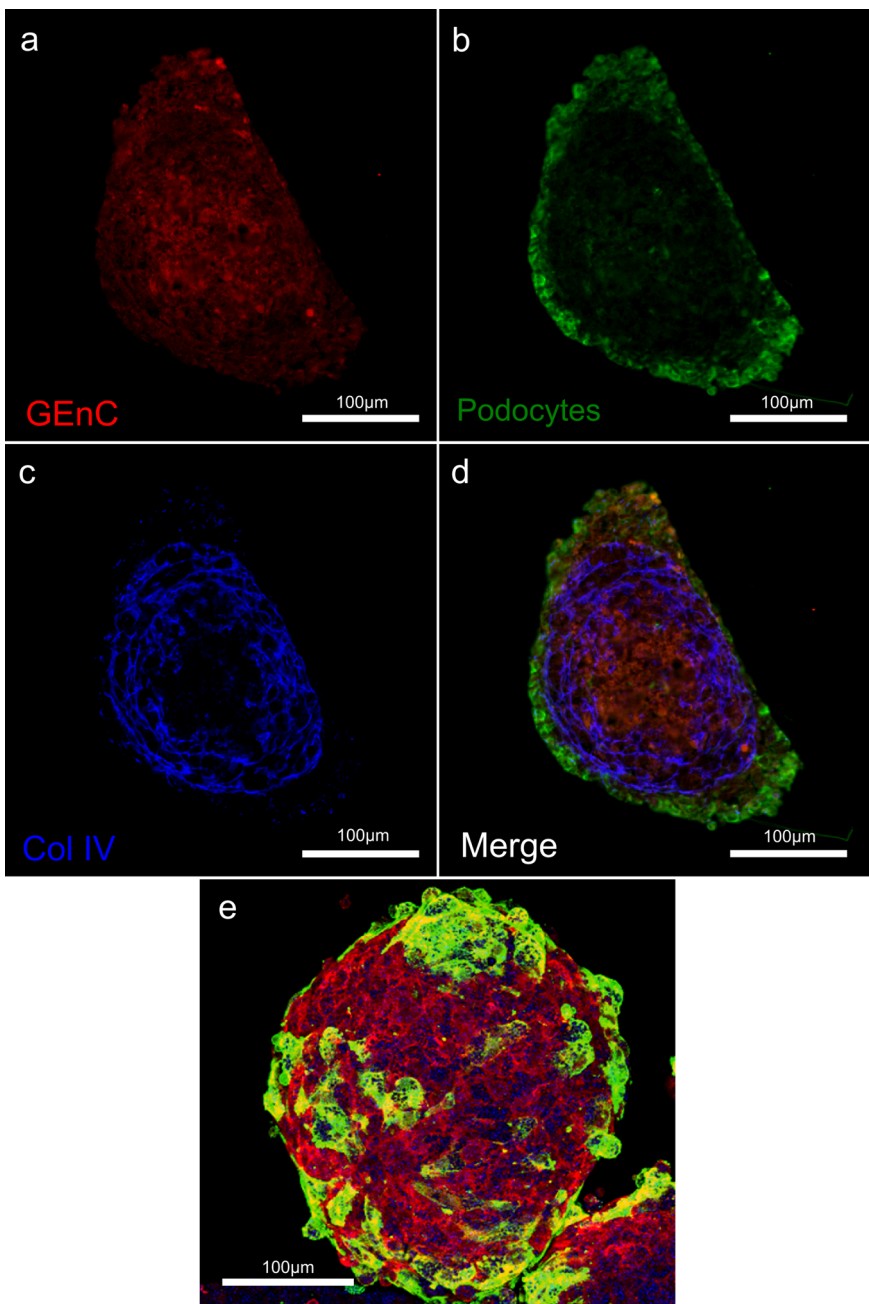

**Fig. 1 Immunofluorescence imaging of GlomSpheres in 2D paraffin sections.** (**a**–**d**) and 3D whole-mount (**e**). **a** Central core of GEnCs (overexpressing M-cherry to aid localisation). **b** Outer layer of podocytes (overexpressing GFP to aid localisation) which wraps the endothelial core. **c** Intermediate deposition of organised collagen IV. **d** Merge, illustrating that collagen IV deposition is concentrated at the podocyte/GEnC interface. **e** Maximum intensity projection of a confocal Z-stack through a GlomSphere. Podocytes (green) appear to interdigitate and are shown to wrap a core of GEnCs (red). Collagen IV staining (blue) shows it to deposited extracellularly. Scale bars = 100 μm.

the immature collagen IV α1 chain was similarly expressed in 2D podocyte, GEnC and co-culture conditions (Fig. 2a, d, g, j). Whilst western blot data suggests a slight increase in collagen IV α1 expression in 2D co-culture conditions, this was not statistically significant (one-way ANOVA $P = 0.2209$) (Fig. 2m). Conversely, the mature collagen IV chain α3 is differentially expressed in podocyte, GEnC and GlomSpheres (Fig. 2b, e, h, k, n). Expression in podocytes and GEnC was shown to be low, with GEnC expression being marginally higher (Tukey's multiple comparison *$P = 0.0408$). GlomSphere α3 expression is significantly higher than both monocultured counterparts (~23 ± 4 and ~24 ± 3 Vs ~44 ± 7) (Tukey's multiple comparison

****$P = < 0.0005$). These findings were validated with western blots (Fig. 2n), where no significant difference between mono-culture conditions was found but GlomSphere α3 expression was found to be significantly higher than podocyte and GEnC spheroids (~16421 ± 2475 VS ~360 ± 186 and ~332 ± 92) (Tukey's multiple comparison ****$P = < 0.0005$). Similarly, the mature α5 subunit of the GBM protein laminin is shown to be absent in mono-cultured podocyte and GEnC conditions but deposited as an organised matrix in GlomSphere (Fig. 2c, f, i, l). Fluorescence intensity quantification showed Laminin α5 expression to be significantly higher in GlomSpheres than in both monocultured conditions (~47 ± 8 Vs ~11 ± .5 and ~14 ± 2)

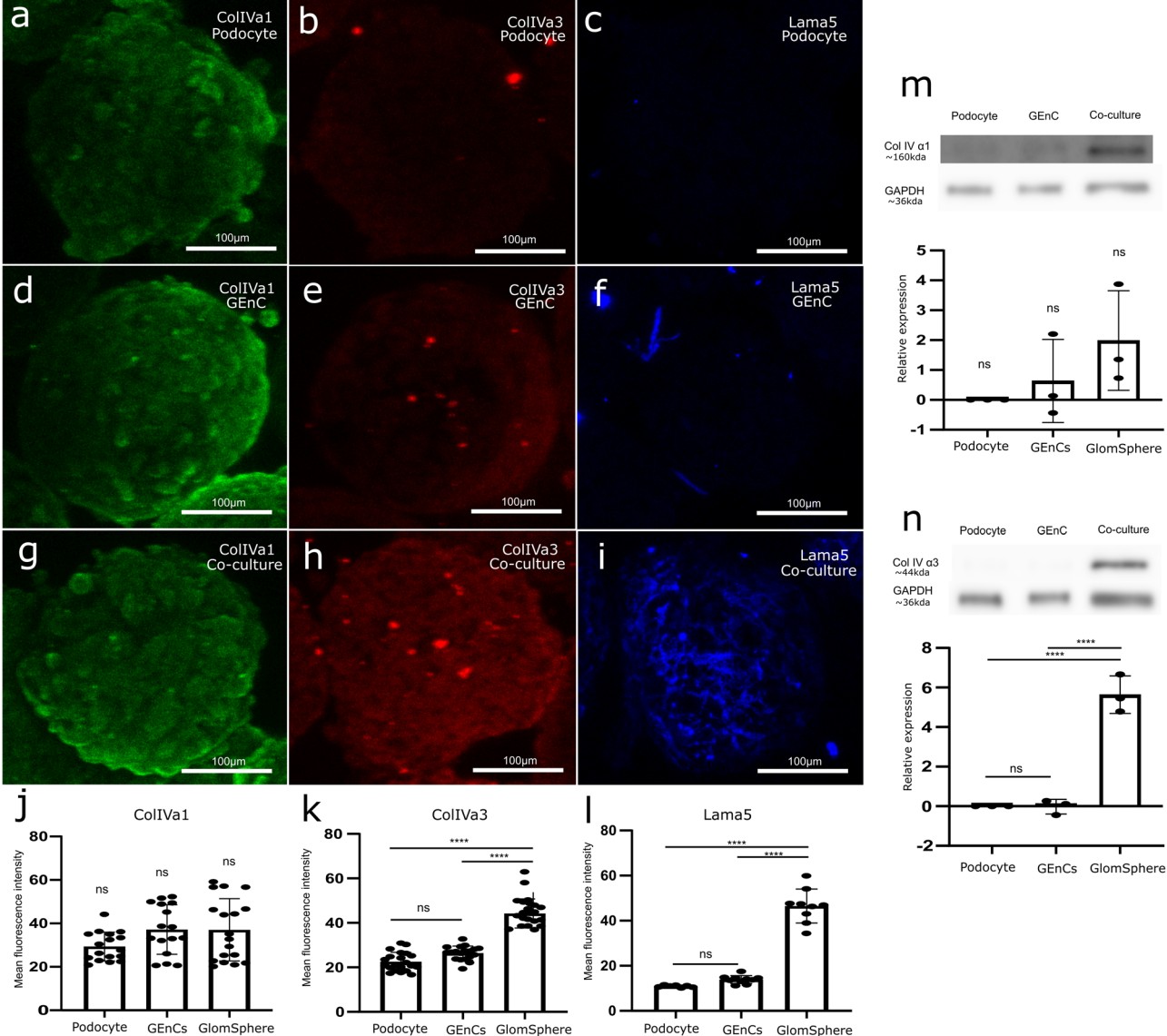

**Fig. 2 Deposition of mature glomerular extracellular matrix proteins in GlomSpheres, compared to podocyte and GEnC mono-culture spheroids.**
**a** Podocyte only spheroid stained for collagen IV α1, which is expressed. **b** Podocyte only spheroid stained for collagen IV α3, which is not strongly expressed. **c** Podocyte only spheroid stained for laminin α5, which is not strongly expressed. **d** GEnC only spheroid stained for collagen IV α1, which is expressed. **e** Endothelial only spheroid stained for collagen IV α3, which is not strongly expressed. **f** GEnC only spheroid stained for laminin α5, which is not strongly expressed. **g** Co-cultured GlomSphere stained for collagen IV α1, which is expressed. **h** Co-cultured GlomSphere stained for collagen IV α3, which has increased expression compared to monocultured spheroids. **i** Co-cultured GlomSphere stained for laminin α5, which is stained more strongly than monocultured spheroids. **j** Mean fluorescence intensity quantification for collagen IV α1 staining (minimum $n = 16$). Mean fluorescence intensity is not significantly different between conditions (one-way ANOVA $p = 0.0814$). **k** Mean fluorescence intensity quantification for collagen IV α3 staining (minimum $n = 19$). GEnC expression was found to be significantly higher than podocytes (Tukey's multiple comparison $*P = 0.0408$). Co-cultured (GlomSphere) expression is shown to be significantly higher than both Podocytes ($****P = < 0.0001$) and GEnCs ($****P = < 0.0001$). **l** Mean fluorescence intensity quantification for laminin α5 staining (minimum $n = 7$). Podocyte and GEnC expression are shown to be insignificantly different (Tukey's multiple comparison $*P = 0.4044$). Co-culture (GlomSphere) expression is significantly higher than Podocytes ($****P = < 0.0001$) and GEnCs ($****P = < 0.0001$). **m** Western blot data for collagen IV α1 expression. Densitometry (minimum $n = 3$) showed no significant difference between podocytes, GEnCs and co-culture (one-way ANOVA $P = 0.2209$). **n** Western blot data for collagen IV α3 expression. Densitometry (minimum $n = 3$) showed no significant difference between podocytes and GEnCs ($P = 0.9985$) Co-culture expression was shown to be significantly higher than both podocytes ($****P = < 0.0001$) and GEnCs.

(Tukey's multiple comparison $****P = < 0.0005$). This antibody is not validated for western blot. Uncropped blots in Supplementary Fig. 7.

**SEM and TEM**. SEM shows GlomSphere surface detail (Fig. 3a–c). The whole GlomSphere image (Fig. 3a) demonstrated the presence of vessel-like protrusions (yellow arrows). These structures appear occasionally from GlomSpheres and stain positive for PECAM-1 (Supplementary Fig. 4). Higher power images (Fig. 3b) reveal what appear to be podocytes inter-digitating (yellow arrows). Higher power images of vessel-like protrusions (yellow arrow) show what appear to be podocytes (red arrows) attached with foot processes to the vessel surface

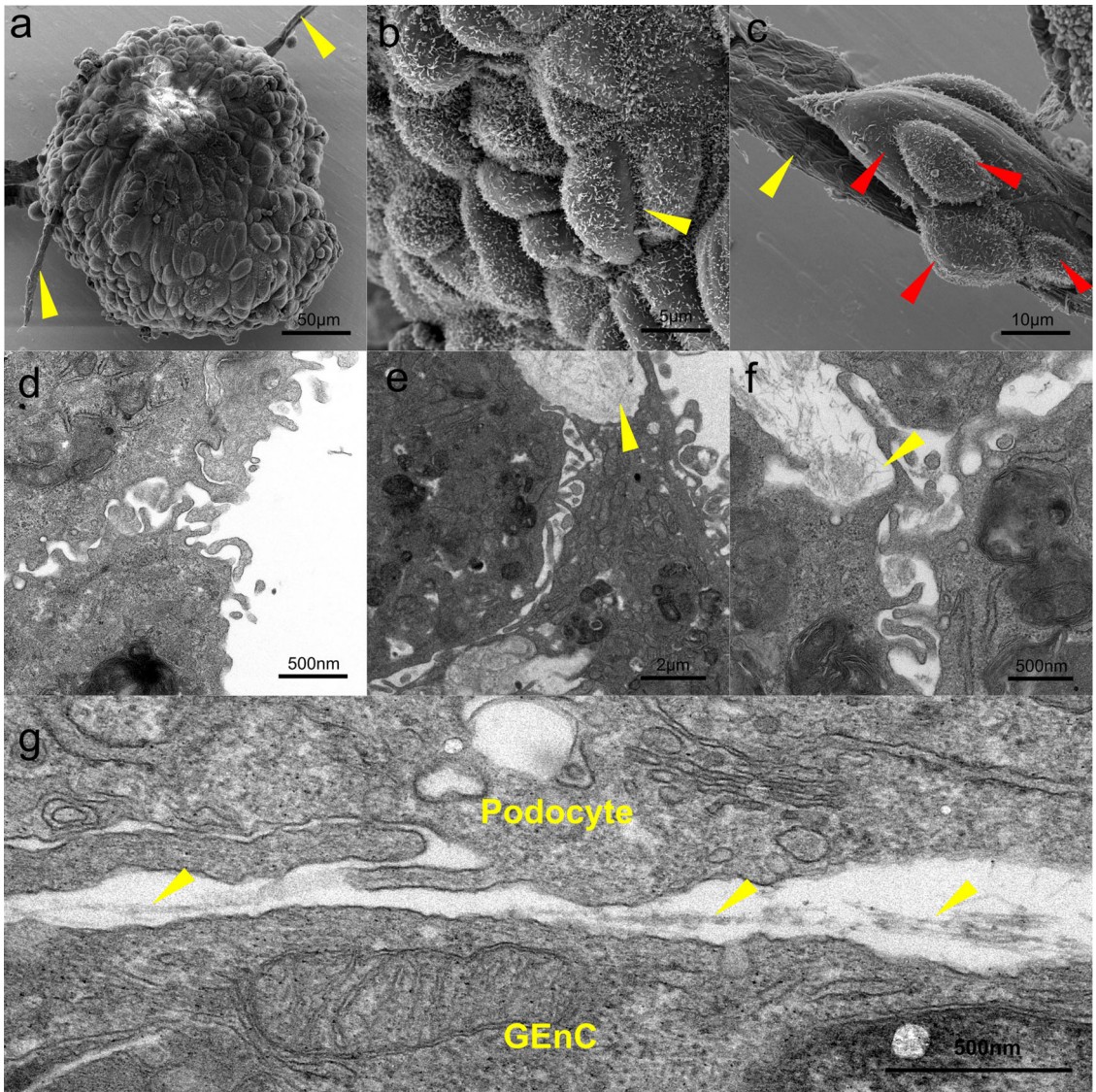

**Fig. 3 Imaging of GlomSphere ultrastructure. a** SEM image of a whole GlomSphere, individual cells are visible on the surface, as are some tubule-like protrusions (yellow arrows). **b** SEM image of podocytes on GlomSphere surface. Short processes coat these cells and appear to interdigitate at cell–cell interfaces (yellow arrow). **c** Podocytes (red arrows) coating the outer surface of a tubule-like protrusion (yellow arrow). The surface of the tubule appears to be coated in podocyte foot-processes. **d** TEM image of podocytes with interdigitating podocytes at GlomSphere surface. **e** TEM image showing interdigitating podocytes, as well as fibrillar collagen deposited into a lumen-like space (yellow arrow). **f** interdigitating podocytes and fibrillar, non-banded collagen deposited in a lumen-like space. **g** A neighbouring podocyte and GEnC deposit a layer of fibrillar collagen IV (yellow arrows) in the extracellular space between them.

(Fig. 3c). TEM of a section through a GlomSphere reveals interdigitation of podocytes at the edges (Fig. 3d). This is further observed in Fig. 3e, f, as well as the presence of fibrillar collagen IV (yellow arrows). A tri-layer structure is shown in Fig. 3g, whereby fibrillar collagen (yellow arrows) is deposited in the extracellular space between a neighbouring podocyte and endothelial cell.

**Upregulation of cell-specific marker proteins**. The expression of key podocyte and GEnC proteins in 2D and GlomSphere samples was examined with human glomerular lysate as a positive control. Western blotting for podocalyxin (which coats podocyte secondary foot-processes) and PECAM-1 (which is located at endothelial cell–cell junctions) was performed (Fig. 4a, b). Whilst podocalyxin appears to be absent in the 2D lysate, it is significantly upregulated in GlomSpheres (~6 fold ± 0.3) (Tukey's

multiple comparison ****$P = < 0.0001$) (Fig. 4a). Compared to the glomerular control, podocalyxin is lower in GlomSpheres (~10 fold increase in Glomerulus VS ~6 fold increase in Glom-Sphere) (Tukey's multiple comparison ****$P = < 0.0001$).

Likewise, whilst PECAM-1 appears to be weakly present in 2D lysates, its expression in GlomSpheres is significantly increased and is approaching the expression level of glomerular lysate (~2.5 fold ± 0.5) (Tukey's multiple comparison ***$P = 0.0006$) (Fig. 4b). The presence of Podocalyxin and PECAM-1 in GlomSpheres was validated using IF staining techniques. Podocalyxin (green) is shown to be present in GlomSpheres and is located primarily at the outer edges, where podocytes are located (Fig. 4c). Under the same illumination settings, podocalyxin staining is negligible in 2D cultured cells (Fig. 4e). Conversely, pecam-1 expression (red) is shown to be present in GlomSpheres but is localised at the core, where the GEnCs are located (Fig. 4d). Staining for PECAM-1 was weakly present in 2D co-cultured cells under the same

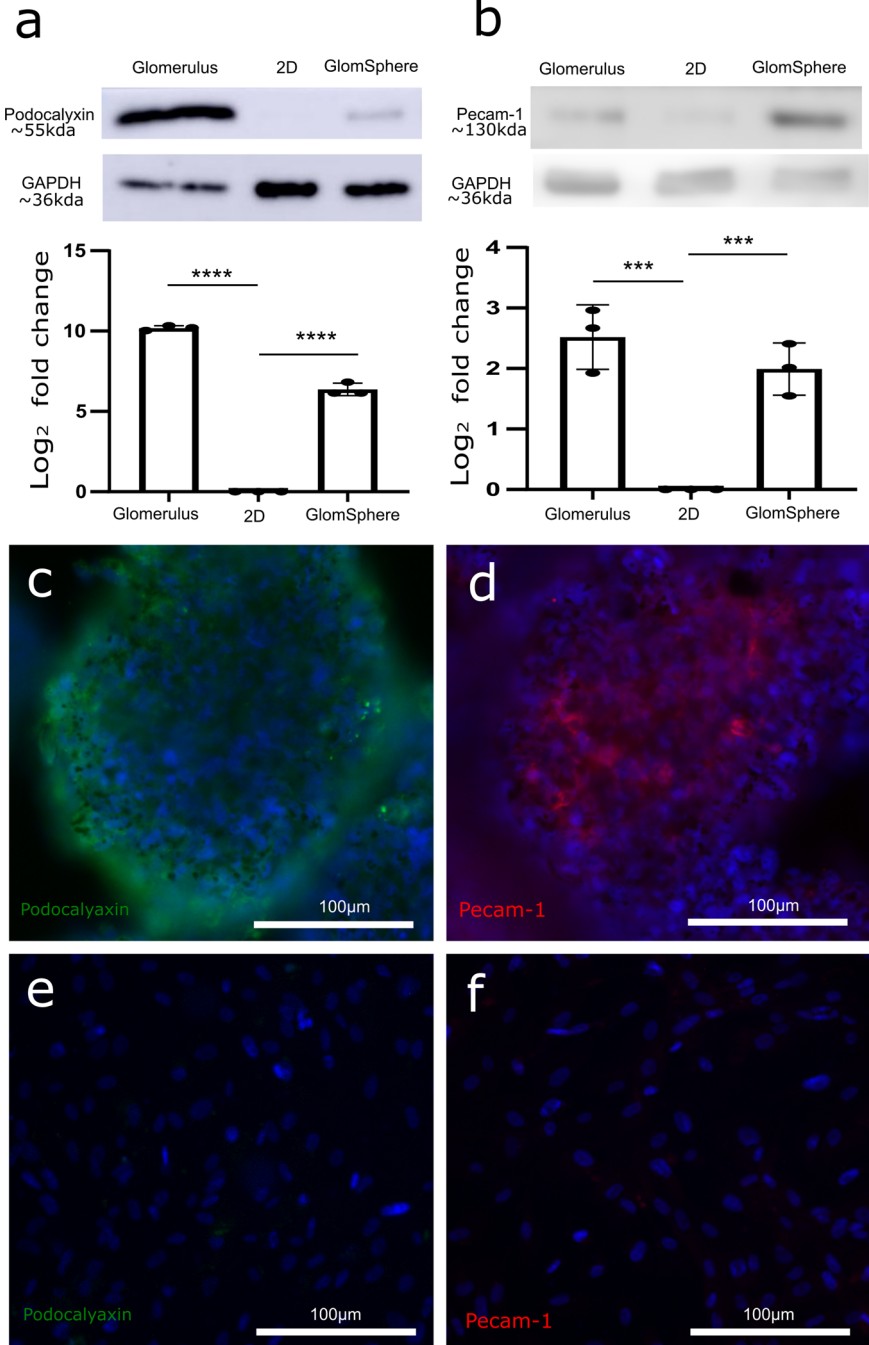

**Fig. 4 Comparative expression of podocalyxin and PECAM-1 in GlomSpheres, 2D cultured cells and human glomeruli. a** Western blot shows podocalyxin expression to be strong in glomerular lysate, absent in the 2D cultured cells and present in GlomSphere lysate. Densitometry confirms (via log2 fold change) that glomerular and GlomSphere podocalyxin expression are significantly higher than 2D cultured cells ($N = 3$) (Tukey's multiple comparison ****$P = < 0.0001$). **b** Western blot shows PECAM-1 expression to be strong in both glomerulus and GlomSphere Lysates, but virtually absent in 2D cultured cell lysate. Densitometry confirms (via log2 fold change) that glomerular and GlomSphere PECAM-1 expression are significantly higher than 2D cultured cells ($N = 3$) (***$P = 0.0006$). **c** Widefield image of a GlomSphere stained for podocalyxin (green) with a DAPI counterstain (blue). Podocalyxin staining is shown to be present and localised primarily to the outer cells in the spheroid, which are podocytes. **d** Widefield image of a GlomSphere stained for PECAM-1 (red) with a DAPI counterstain (blue). PECAM-1 staining is present and localised primarily to the cells at the core of the GlomSphere, which are GEnCs. **e** Widefield image of 2D co-cultured podocytes and GEnCs stained for podocalyxin (green) with a DAPI counterstain (blue). Whilst the same exposure/gain settings as GlomSpheres were used, podocalyxin staining is virtually absent. **f** Widefield image of 2D co-cultured podocytes and GEnCs stained for PECAM-1 (red) with a DAPI counterstain (blue). Again, the same exposure/gain settings as GlomSpheres were used and Pecam-1 staining is only very weakly present.

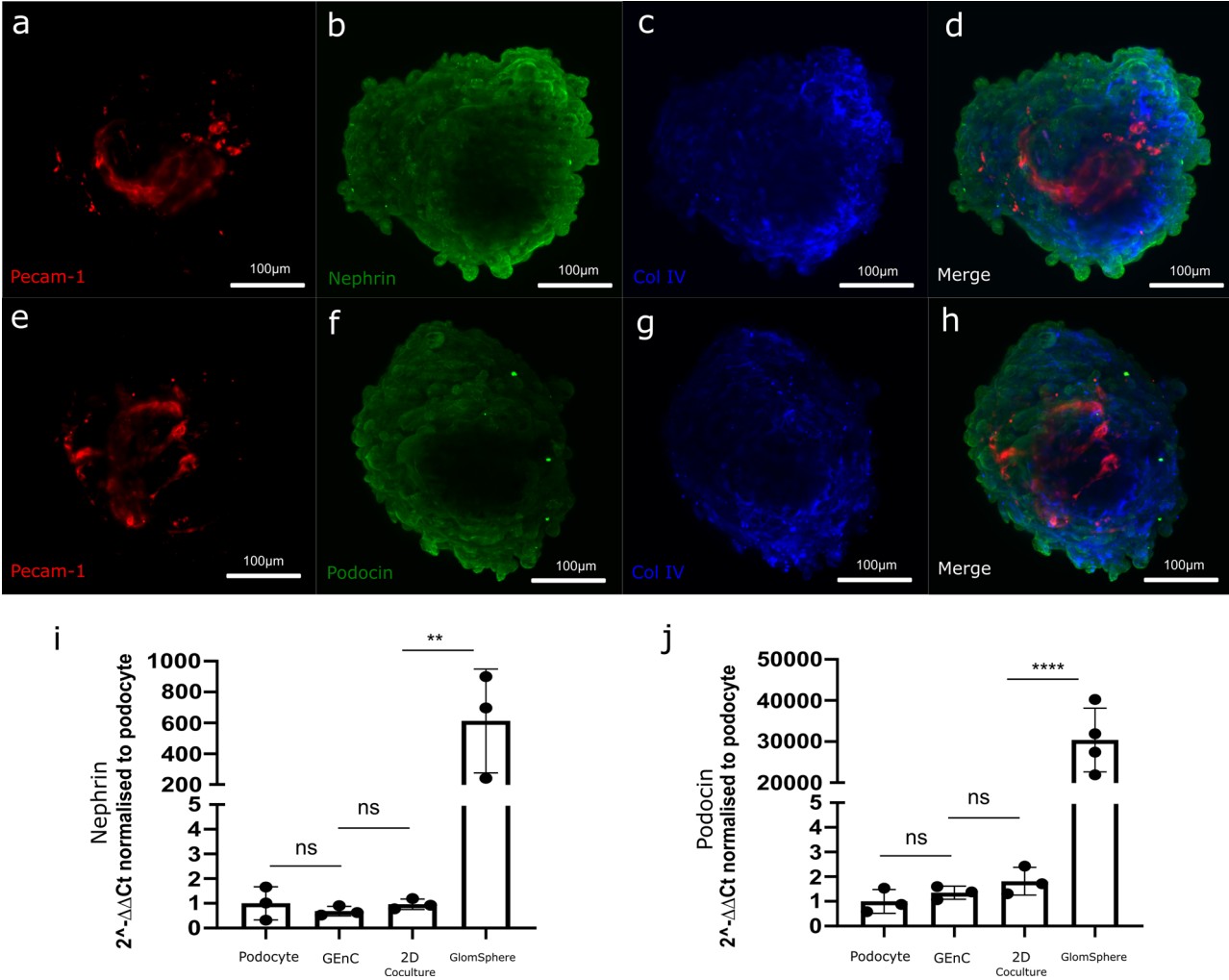

**Fig. 5 Lightsheet microscopy images of GlomSpheres expressing key glomerular cell markers. a** Organised pecam-1 staining is localised to the centre of the GlomSphere, where GEnCs are located. **b** Nephrin staining on GlomSphere periphery, where podocytes are located. **c** Collagen IV staining is extracellular and is strongest in the region between Nephrin and Pecam-1 staining. **d** Merge. **e** Pecam-1 staining is once again shown to be organised and central in location. **f** Podocin staining on GlomSphere periphery is located primarily where podocytes are located. **g** Collagen IV staining is once again concentrated at the areas between podocytes and GEnCs. **h** Merge. **i** QPCR results for nephrin. $2^{-\Delta\Delta Ct}$ results are shown relative to 2D podocytes and whilst there is no significant difference between 2D conditions, GlomSphere $2^{-\Delta\Delta Ct}$ is significantly elevated relative to 2D cocultured cells (Tukey's multiple comparison **$P = 0.0092$). **j** QPCR results for podocin. $2^{-\Delta\Delta Ct}$ results are shown relative to 2D podocytes and whilst there is no significant difference between 2D conditions, GlomSphere $2^{-\Delta\Delta Ct}$ is significantly elevated relative to 2D cocultured cells (Tukey's multiple comparison ****$P = < 0.0001$).

illumination. Collectively, these results indicate that 2D culture results in diminished podocyte and GEnC marker expression and that GlomSphere culture is restorative. Uncropped blots in Supplementary Fig. 7.

The presence of key podocyte markers podocin and nephrin was then examined. Wholemount triple-IF staining for PECAM-1, nephrin and collagen IV (Fig. 5a–d) demonstrates an organised central core of PECAM-1 staining, surrounded by a mostly peripheral nephrin stain. collagen IV staining is concentrated primarily at the area between PECAM-1 and nephrin. This staining pattern is repeated for the PECAM-1, podocin and collagen IV triple stain (Fig. 5e–h), with podocin staining located peripherally. The intricate, vessel-like organisation of PECAM-1 staining can be more clearly observed (in Supplementary Movie 1), whereby a branching vascular network is at the core of GlomSpheres.

QPCR was employed to examine nephrin and podocin mRNA levels (Fig. 5i, j, respectively). It is apparent that whilst there is

minimal expression and very little difference in nephrin mRNA between each of the 2D culture conditions, there is a large and significant increase when podocytes and GEnCs are cultured as GlomSpheres (600 fold increase, Tukey's multiple comparison **$P = 0.0092$) (Fig. 5i). The same is true for podocin mRNA, which is also not significantly different between 2D cultures, but there is an even greater increase in GlomSphere cultures (30,000 fold increase, Tukey's multiple comparison ****$P = < 0.0001$) (Fig. 5j). A comparison of GlomSphere and human kidney cortex expression of podocin and nephrin was also made as a positive control for primers (Supplementary Fig. 6). Despite variability between cortex samples (likely due to differing glomeruli counts), expression levels were stronger than GlomSpheres (more so for nephrin than podocin, which was closer to in vivo expression).

**Induction and attenuation of disease phenotypes in GlomSpheres.** A series of experiments was then undertaken to manipulate disease phenotypes in GlomSpheres. Initially, GlomSpheres were cultured

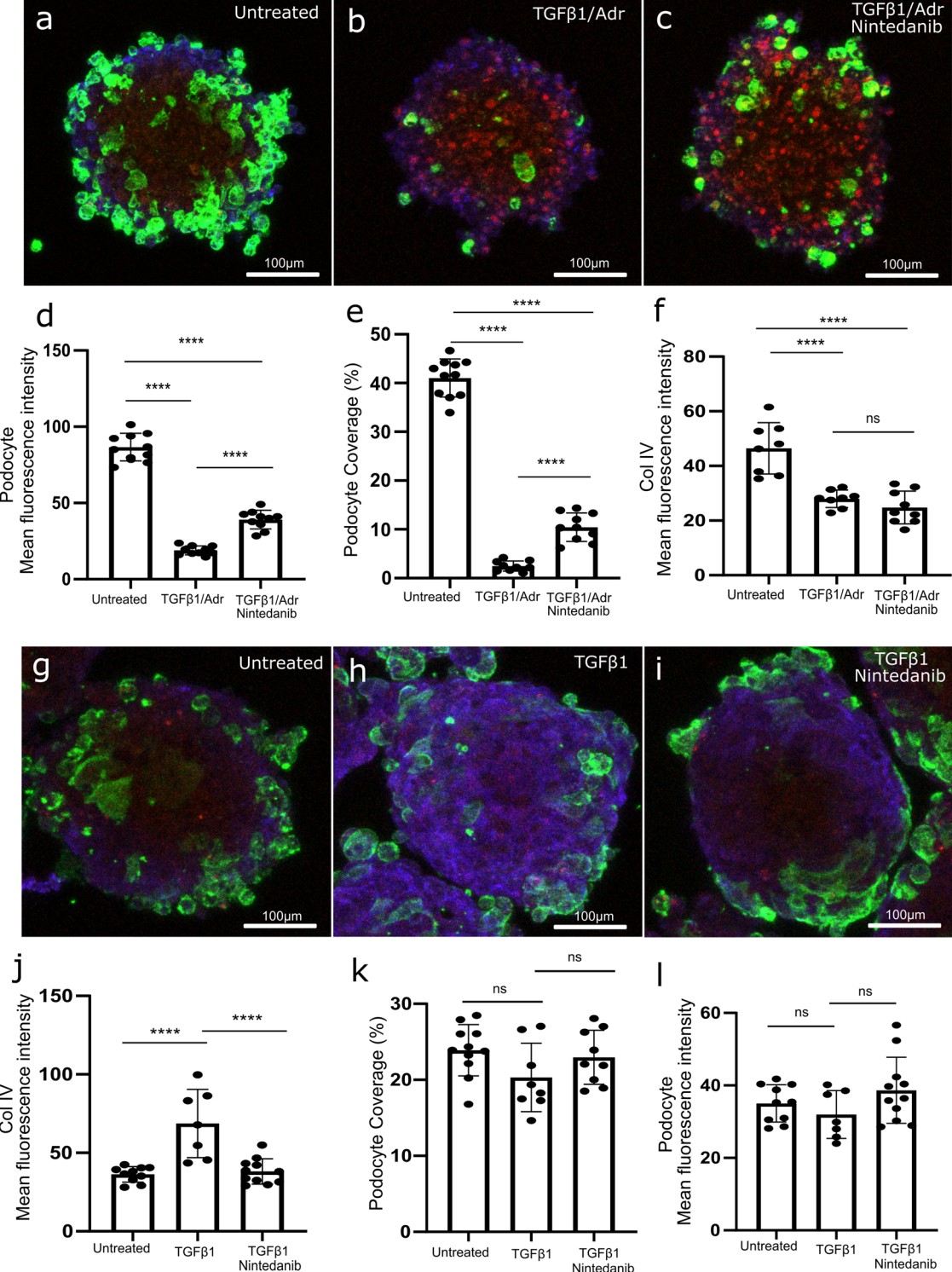

in the presence of profibrotic agent TGF-β1, with or without Adriamycin, a reno-toxic agent known to induce a focal segmental glomerulosclerosis (FSGS) phenotype in rat models[18,19] (Fig. 6a–f). GlomSphere incubation with 10 ng/ml TGF-β1 and 10 nM Adriamycin for 72 h was shown to induce dramatic podocyte detachment from the spheroid's surface compared to untreated GlomSpheres (Fig. 6a, b). A clarification of podocyte detachment is shown in Supplementary Fig. 5. Quantification of podocyte (GFP) mean fluorescence intensity confirms a statistically significant loss of podocyte fluorescence after TGF-β1 and Adriamycin treatment (~87 ± 9 Vs ~19 ± 3, respectively) (Tukey's multiple comparison

****$P = < 0.0005$) (Fig. 6d). This loss is also observed when quantifying podocyte percentage coverage (~41% Vs ~2.5%, respectively). Nintedanib is an inhibitor of platelet-derived growth factor receptor, fibroblast growth factor receptor and vascular endothelial growth factor and is a licenced treatment for idiopathic pulmonary fibrosis (IPF) with antifibrotic properties[20–22]. The addition of 5 nM Nintedanib at the same time as the TGF-β1 and Adriamycin treatment reduced the loss of podocytes from the surface of the spheroid (Fig. 6c). Quantification of podocyte (GFP) mean fluorescence intensity and podocyte coverage confirms a statistically significant retention of podocyte fluorescence compared to TGF-β1

**Fig. 6 Induction and modulation of fibrotic phenotypes in GlomSpheres. a** Max-intensity projection image of an untreated GlomSphere. Podocytes (green) are shown to peripherally wrap a GEnC core (red). Collagen IV is stained (blue) and is located primarily at the podocyte/GEnC interface. **b** GlomSphere after 72 h incubation with TGF-β1 (10 ng/ml) and Adriamycin (10 nM). Podocyte coverage on the surface of GlomSphere is discernibly reduced. Collagen IV fluorescence is also possibly reduced. **c** GlomSphere after incubation with TGF-β1 (10 ng/ml) and Adriamycin (10 nM) as well as Nintedanib (10 μM). Whilst podocyte coverage appears diminished relative to the untreated GlomSphere, there are discernibly more podocytes attached than the TGF-β1/Adriamycin condition. **d** Quantification of GFP (podocyte) mean fluorescence intensity reveals that untreated GFP fluorescence is significantly higher than TGF-β1/Adriamycin and Nintedanib conditions (Tukey's multiple comparison $N = > 8$, ****$P = < 0.0001$). The addition of Nintedanib significantly increased GFP fluorescence relative to the TGF-β1/Adriamycin condition (****$P = < 0.0001$). **e** Quantification of podocyte coverage percentage confirms that untreated GFP fluorescence is significantly higher than TGF-β1/Adriamycin and Nintedanib conditions (Tukey's multiple comparison $N = > 8$, ****$P = < 0.0001$). The addition of Nintedanib significantly increased GFP fluorescence relative to the TGF-β1/Adriamycin condition (****$P = < 0.0001$). **f** Quantification of collagen IV mean fluorescence intensity reveals it to be significantly decreased in TGF-β1/Adriamycin and Nintedanib conditions relative to the untreated control ($N = > 8$) (****$P = < 0.0001$). **g** Untreated GlomSphere. **h** GlomSphere after 72 h incubation with TGF-β1 (10 ng/ml) only. Collagen IV (blue) expression appears to have increased and is no longer primarily at the podocyte/GEnC interface. There is no discernible difference in podocyte (green) coverage. **i** GlomSphere after 72 h incubation with TGF-β1 (10 ng/ml) and Nintedanib (10 μM). Collagen IV (blue) fluorescence is reduced relative to the TGF-β1 only condition. **j** Quantification of collagen IV mean fluorescence intensity reveals it to be significantly increased in TGF-β1 relative to the untreated control and Nintedanib conditions ($N = > 7$) (***$P = 0.0006$ and ***$P = 0.0003$ respectively). **k** Quantification of GFP (podocyte) mean fluorescence intensity. There is no significant difference between conditions. **l** Quantification of podocyte coverage percentage. There is no significant difference between conditions.

and Adriamycin only (~19 ± 3 TGF-β1 and Adriamycin Vs ~39 ± 6 with the addition of Nintedanib) (Tukey's multiple comparison ****$P = < 0.0005$) (Fig. 6d) and (~2.5% Vs ~10.5%) (Fig. 6e). Interestingly, collagen IV fluorescence appears to have been lost in the TGF-β1/Adriamycin and Nintedanib conditions (Fig. 6f).

These findings led to the hypothesis that Adriamycin-mediated podocyte loss was occurring before excessive production of collagen IV could occur. To test this, TGF-β1 (without Adriamycin) was administered to GlomSpheres for 72 h (Fig. 6g–i). It is apparent that TGF-β1 incubation induced a dramatic increase in collagen IV deposition when compared to the untreated control (Fig. 6g, h). This is confirmed by quantification of collagen IV mean fluorescence intensity, which shows a statistically significant increase in fluorescence after TGF-β1 treatment (~36 ± 5 Vs ~69 ± 22 Tukey's multiple comparison ***$P = 0.0003$) (Fig. 6h). This increase in collagen IV fluorescence appears to have been attenuated with the addition of Nintedanib (~69 ± 22 Vs 38 ± 8) (Fig. 6i, j) (Tukey's multiple comparison ***$P = 0.0006$). Interestingly, podocyte loss appears to have been negligible across all three conditions (Fig. 6g, h, i, k, l). Jointly, these experiments highlight two capabilities of the GlomSphere model, one of which models early podocyte loss (TGF-β1 and Adriamycin), the other modelling fibrotic ECM dysregulation (TGF-β1 only).

**GlomSphere incubation with nephrotic syndrome (NS) patient plasma.** Whilst an artificially induced disease phenotype is a powerful tool, a greater test of a GlomSphere's ability to model glomerular disease is to examine its response to nephrotic syndrome (NS) patient plasma. The samples were from plasma exchange effluent of patients suffering immediate recurrence of their disease post-renal transplantation (with institutional ethical permissions)[14]. GlomSpheres were incubated with either NS relapse, remission, or relapse + Nintedanib human plasma samples for 7 days (Fig. 7). Podocyte loss was first examined (Fig. 7a–c, g, h). It is apparent that podocyte coverage is weaker in relapse Glom-Spheres (Fig. 7a) relative to remission (Fig. 7b) and relapse + Nintedanib conditions (Fig. 7c). Quantification of GFP fluorescence intensity and podocyte coverage confirms that relapse plasma significantly reduces podocyte attachment (intensity ~9 ± 1 Vs ~17 ± 2) (Tukey's multiple comparison ****$P = < 0.0001$) (podocyte coverage ~29% Vs ~57% (Tukey's multiple comparison ****$P = < 0.0001$). It is also clear that the addition of Nintedanib attenuates this loss significantly when compared to relapse only (intensity ~9 ± 1 Relapse Vs 13 ± 2 Relapse + Nintedanib) (Tukey's

multiple comparison ****$P = < 0.0001$) (podocyte coverage ~29% Relapse Vs ~46% Relapse + Nintedanib) (Tukey's multiple comparison ****$P = < 0.0001$). Collagen IV deposition also appears to increase in relapse conditions, relative to remission (Fig. 7d, e). Relapse + Nintedanib collagen IV appears to have a different staining pattern (Fig. 7d, f). Quantification of collagen IV mean fluorescence intensity confirms that relapse is significantly higher than remission (~100 ± 20 Vs ~61 ± 15 Tukey's multiple comparison (**$P = < 0.0014$). Collagen IV fluorescence is shown to be significantly higher in the relapse + Nintenadib condition relative to both relapse and remission (~129 ± 51 Vs ~100 ± 20 and ~61 ± 15, respectively, Tukey's multiple comparison *$P = 0.0401$ and ****$P = < 0.0001$, respectively). This indicates that whilst Ninte-danib appears to change the collagen IV deposition pattern, it is not able to attenuate dysregulation and may exacerbate it.

**A rapid pharmaceutical screening approach.** The nature of spheroid culture lends itself well to higher throughput screens. To demonstrate the ability of GlomSpheres to run a rapid pharmacological screen, a rapid imaging approach was employed using the In-cell analyser. A podocyte cell-line generated from a patient with a pathogenic R138Q mutation of podocin[14] was used for these experiments to study the ability of various compounds to rescue podocin functionality. Podocyte loss from the spheroid was quantified as a surrogate for podocyte adhesiveness (which is a well-established readout of podocyte pathology in in vitro assays) and was compared in wildtype (WT), podocin mutant (PM) and PM cells treated with various compounds. These compounds are derived from a previously screened NCBI library, where a single compound (termed 407) has been found to rescue podocin functionality both in vitro and in vivo. The current experiment tested this compound against chemically-modified variants to assess relative podocyte phenotype rescue efficacy. After 5 days of treatment, podocyte attachment (via GFP expression) was compared (Fig. 8).

Firstly, it is shown that wildtype podocytes (Fig. 8b) attach more strongly than podocin mutant podocytes (Fig. 8a). This is confirmed by quantification of GFP mean fluorescence intensity within spheroid area (Fig. 8d) and podocyte coverage percentage (Fig. 8e). Secondly, the ability of a suite of compounds to restore podocyte attachment to that of wildtype podocytes is also shown (one of which is shown in Fig. 8c). This is confirmed through quantifications, which also demonstrate the ineffectiveness of other compounds (Fig. 8d, e).

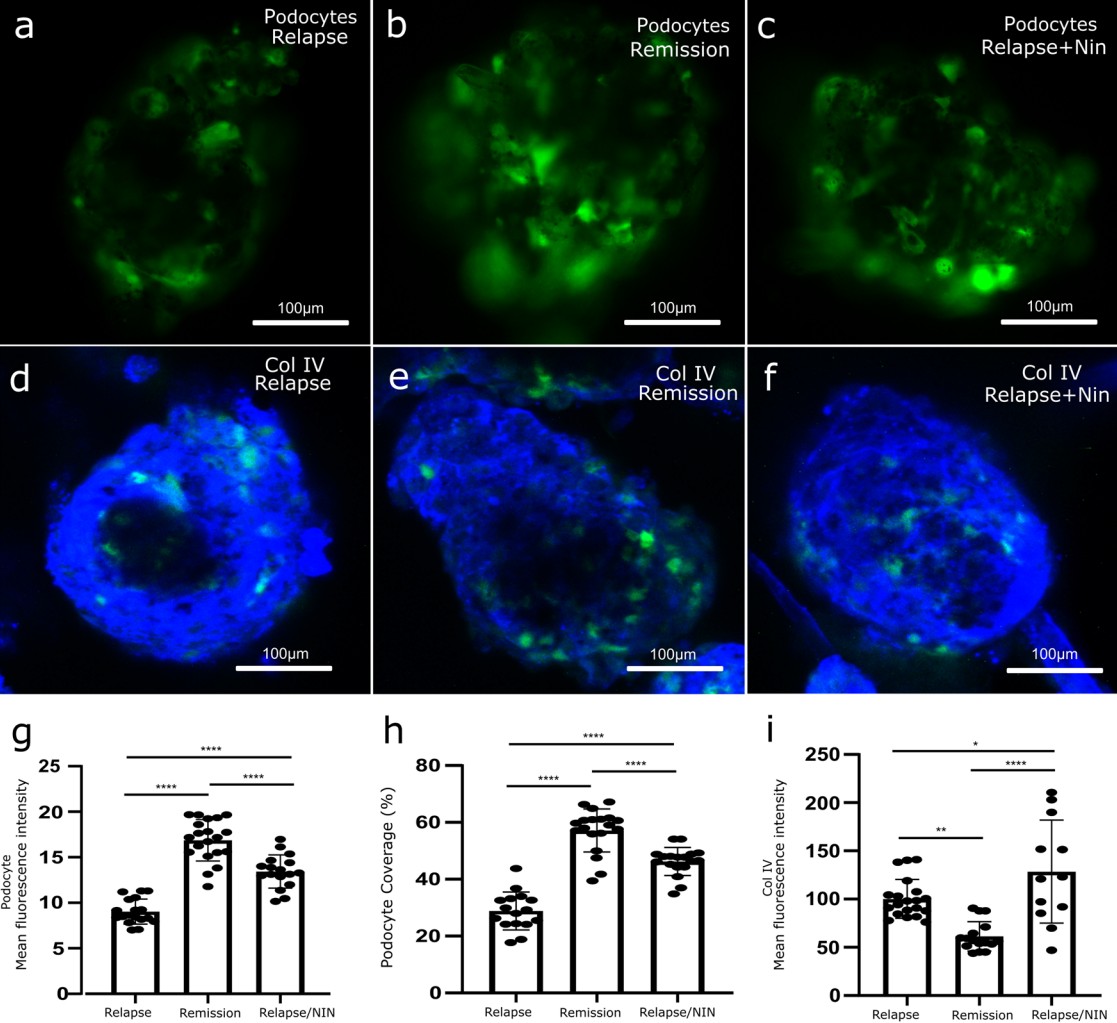

**Fig. 7 NS patient plasma treated GlomSpheres. a** GlomSphere podocytes (green) after 7 days incubation with a relapsed CKD patient plasma.
**b** GlomSphere podocytes (green) after 7 days incubation with a remission CKD patient plasma. There is visibly more podocyte fluorescence than in the relapsed condition. **c** GlomSphere podocytes (green) after 7 days incubation with relapsed CKD patient plasma with 10 nM Nintedanib. There is visibly more podocyte fluorescence than with relapse plasma alone. **d** GlomSphere stained for collagen IV (blue) after 7 days incubation with a relapsed CKD patient plasma. **e** GlomSphere stained for collagen IV (blue) after 7 days incubation with a remission CKD patient plasma. There is visibly less collagen IV fluorescence than in the relapsed condition. **f** GlomSphere stained for collagen IV (blue) after 7 days incubation with relapse CKD patient plasma with 10 nM Nintedanib. Whilst the staining pattern is more similar to the remission sample, the intensity appears to be more akin to the relapse sample.
**g** Quantification of podocyte GFP mean fluorescence intensity. Relapse ($N = 18$) fluorescence is significantly lower than remission ($N = 17$) and relapse/Nin ($N = 20$) (Tukey's multiple comparison (****$P = < 0.0001$)). Remission fluorescence is significantly higher than relapse/Nin (****$P = < 0.0001$).
**h** Quantification of podocyte coverage percentage. Relapse ($N = 18$) fluorescence is significantly lower than remission ($N = 17$) and relapse/Nin ($N = 20$) (Tukey's multiple comparison (****$P = < 0.0001$)). Remission fluorescence is significantly higher than relapse/Nin (****$P = < 0.0001$). **i** Quantification of collagen IV mean fluorescence intensity. Relapse ($N = 20$) fluorescence is significantly higher than remission ($N = 12$) (Tukey's multiple comparison (**$P = < 0.0014$)) but significantly lower than relapse/Nin ($N = 16$) (*$P = 0.0401$). Relapse/Nin collagen fluorescence is also higher than remission (****$P = < 0.0001$).

A standard podocyte adhesion assay was performed to compare GlomSphere results to a more traditional 2D, monoculture assay of cell attachment (Fig. 8f). These results suggest the GlomSphere assay has greater dynamic range than a 2D assay. The broad direction of results between the two assays is similar for each compound (with some notable exceptions such as 606) but the range of differences is extended in the 3D assay, with more compounds in the latter therefore reaching statistical significance.

## Discussion
The data presented here demonstrate an effective method of enhancing the phenotype of conditionally immortalised human podocytes and GEnCs in 3D. It is apparent that the superior environment presented by spheroid culture induces an organised, glomerular basement membrane (GBM)-like extracellular matrix at the podocyte/GEnC interface. Importantly, we have demonstrated that by co-culturing podocytes and GEnCS in spheroids, mature collagen IV α3 and laminin α5 GBM proteins are dramatically upregulated. This has never previously been achieved in 2D culture models[23]. This is reminiscent of the developmental switch that happens in vivo, whereby glomerular isoforms of collagen IV transition from predominantly immature COL4α1α2α1 networks to mature COL4α3α4α5 networks[24]. Likewise, during glomerulogenesis laminin-111 α1β1γ1 is replaced by mature laminin-521 α5β2γ1[25].

The interdigitating foot processes seen in TEM and SEM images (Fig. 3) are further indicative of a glomerulus-like

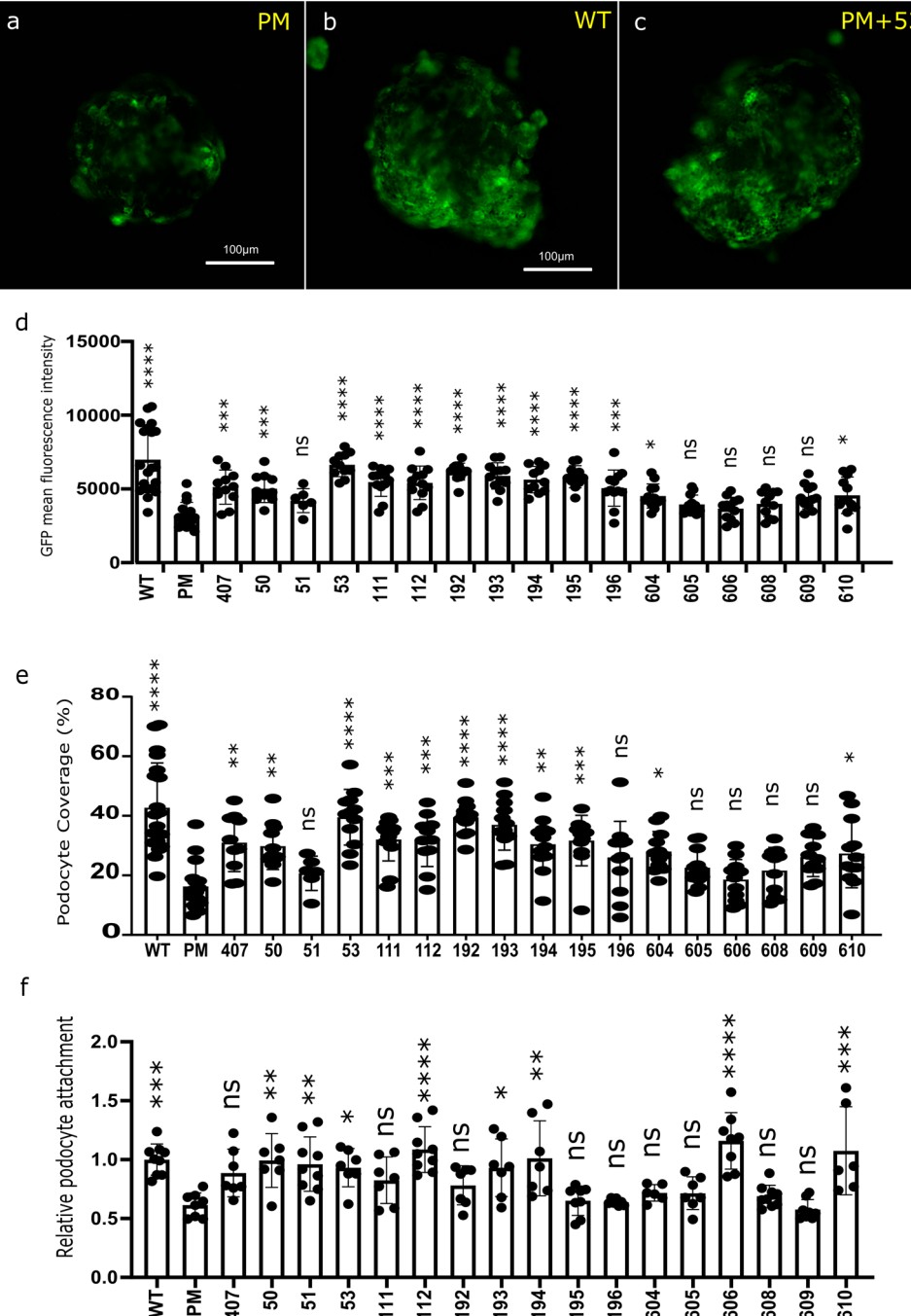

**Fig. 8 High throughput screening of pharmaceuticals using InCell analyser with GlomSpheres. a** Untreated LP (podocin mutant) GlomSphere. Podocytes (green) are shown to incompletely cover GlomSphere surface. **b** Untreated LY (wild type) GlomSphere. Podocytes more completely cover GlomSphere surface, increasing its apparent size. **c** LP (podocin mutant) spheroid treated for 5 days with a podocin-modifying compound (53). Podocyte retention appears to have been improved and the GlomSphere looks similar to the wild type condition. **d** Quantified GFP mean fluorescence intensity of GlomSpheres treated with several disease modifying compounds. Many of these compounds are sufficient to restore GFP fluorescence (and therefore podocyte retention) to wildtype (LY) levels when compared to untreated GlomSpheres (Un). **e** Quantified podocyte coverage percentage of GlomSphere following treatment. Once again the assay appears to have demonstrated several compounds capacity to protect podocyte retention, whilst showing that others are ineffective. **f** 2D podocyte adhesion assay testing the same compounds. Results shown as relative podocyte attachment. It is shown that whilst WT and PM adhesion is significantly different, many compounds appear less effective at rescuing adhesion when tested with this assay. Statistical analyses are Tukey's multiple comparison. ****refers to $P = < 0.0001$, ***refers to $P = > 0.0001$, **refers to $P = > 0.001$, *refers to $P = > 0.01$. (~$N = 12$ spheroids per condition).

maturation of podocytes in GlomSpheres. The ability of podocytes to interdigitate under GlomSphere culture may also be underpinned by the upregulation of the critical slit diaphragm proteins nephrin and podocin (Fig. 5) and suggest more physiological slit diaphragm signalling.

The vessel-like protrusions seen in Fig. 3 are suggestive of glomerulus like maturation in GEnCs, especially as they appear to be PECAM-1 positive (Supplementary Fig. 4). Figure 5 and Supplementary Movie 1 demonstrate that this network of PECAM-1 expressing vessels is present throughout the centre of

GlomSpheres and Fig. 4 indicates that whilst PECAM-1 expression is extremely low in 2D cultured cells, in GlomSpheres it is returned to almost glomerular levels. These results are interesting when compared to the findings of Rinschen et al.[9] which demonstrated low levels of glomerular marker proteins from 2D monocultured conditionally immortalised (CI) podocytes. Our data suggest that 3D co-culture of CI GEnCs and podocytes as part of GlomSpheres may have a restorative effect.

Experiments that artificially modulate disease phenotypes in GlomSpheres display their potential for pharmaceutical screening. TGFβ1 and Adriamycin are known drivers of fibrosis[26–28] and as hypothesized, their combined application induced podocyte loss in GlomSpheres (Fig. 6). Nintedanib has previously been shown to interfere with fibroblast proliferation and differentiation and extracellular matrix deposition[29,30]. It was hypothesized that the addition of Nintedanib to the fibrotic milieu would attenuate the induced fibrotic phenotype somewhat and this is apparent in Fig. 6, where podocyte loss is reduced with Nintedanib addition.

Adriamycin is known to induce ECM remodelling but also highly toxic to cells[31,32]. It was hypothesized that podocytes were being lost from GlomSpheres before ECM remodelling could occur. TGF-β1 was therefore applied to GlomSpheres as the sole fibrotic stimulus and as predicted, ECM dysregulation was observed (Fig. 6). This dysregulation is also attenuated by the antifibrotic properties of Nintedanib. Taken together, these fibrotic modulation experiments demonstrate the ability of GlomSpheres to model two key aspects of glomerulosclerosis, podocyte loss and ECM dysregulation, and to examine them separately. Moreover, the demonstrated attenuation properties of Nintedanib demonstrate GlomSphere's potential as platform for drug-screening.

Incubation with NS patient plasma samples allows for an assessment of GlomSphere's response to a disease milieu, which will contain disease-altering circulating factors other than TGF-β1[14]. As hypothesized, FSGS relapse plasma was shown to induce both podocyte loss and an increase in collagen IV deposition, relative to remission plasma (Fig. 7), which is distinct from the effects of TGFβ1or Adriamycin, in this system. Nintedanib appears to have a similar protective effect on podocyte loss in FSGS relapse plasma to that observed with TGFβ1/Adriamycin conditions (Fig. 6 vs Fig. 7). This protective effect is however not observed with regards to collagen IV deposition, highlighting the limits of Nintedanib treatment in this case. Whilst it would be optimal to compare relapse and remission patients to a normal control patient, healthy patients do not undergo plasmapheresis like FSGS patients. For this reason, the internal control offered by examining the same patient as different dates (in relapse and remission) is the fairest control.

The use of the In cell analyser (Fig. 8) to conduct a higher throughput drug screen demonstrates the rapidity and scalability of GlomSpheres. This assay was performed on live cells and imaging an entire 96-well plate takes <5 min. Quantification can also be performed automatically on the system, removing human error and increasing throughput. This experiment was limited to 12–18 spheroids per condition due to limited quantities of precious compounds. There is however no reason an assay of this nature cannot be upscaled considerably. Interestingly, comparison with the 2D adhesion assay shows the greater dynamic range of the GlomSphere assay. One of these was the compound termed "407", which all other compounds are variants of. This compound has been shown to be effective in animal studies[33] and its increased potency in the GlomSphere assay suggests increased physiological relevance of this model.

There are limitations with the GlomSphere model, including the lack of glomerular mesangial cells, which can contribute to the signalling milieu as well as mesangial derived matrix

components, and structural support for capillaries[34]. The latter is unlikely to be adequately recreated unless flow is also provided, which is currently beyond the capabilities of this type of model, though it was interesting that primitive vascularisation is seen in GlomSpheres, indicating that enough soluble signals (e.g. VEGF) are present to initiate capillary formation. The GlomSphere model is also currently not a model of glomerular function (i.e. does not model filtration efficiency). We are currently exploring possibilities to modify the model to this end, although scalability is likely to be compromised in this case.

3D culture is emerging as an essential tool to span the gap between 2D culture and animal models, with a far greater ability to mimic the behaviour of in vivo tissue. Practical limitations to date include the need to incorporate pluripotent stem cells (in some systems) or external matrices, reproducibility, expense, and complexity particularly if requiring technical scaffolds. These shortcomings have resulted in very little change to the industrial drug-discovery pipeline, which is still reliant on over-simplistic 2D cell models. We have developed technique that overcomes both the problem of representativeness and scalability/reproducibility, as well as being extremely straightforward to analyse.

Collectively, our data demonstrate that GlomSpheres represent an excellent method of rapidly and reproducibly 3D co-culturing glomerular cell lines. More broadly this method shows the potential to co-culture mature differentiated cells in a way that more precisely recreates 3D tissue properties. There is an urgent need for effective treatments for glomerular diseases, and for antifibrotic drugs, which are more likely to be discovered using a model that accurately simulates fibrosis. To the best of our knowledge, GlomSpheres represent a new in vitro standard, as a scalable, 3D model of the glomerulus with great potential as a model for drug-screening.

## Methods

**Cell lines and cell culture**. Conditionally immortalised podocytes and GEnCs were isolated and cultured as previously described[6,7]. When required, these cell lines were further transfected to generate stable GFP-actin overexpressing podocytes and mCherry-actin overexpressing GEnCs as previously described[34]. Podocin-mutant (PM) cells were used as previously described[14]. Each individual cell line is from a different single patient, and as such, are from different genetic backgrounds. Several clones of each are taken and used interchangeably in our lab. No difference between clones is observed. For all co-culture experiments, cells were cultured in EBM-basal culture medium (Lonza) supplemented with Bulletkit growth factors to produce EGM-2 MV growth medium (Lonza). During proliferation, cells were cultured at 33 °C, before being differentiated at 37 °C for 10 days. '2D co-culture' conditions in experiments are achieved by mixing podocytes and GEnCs at a 1:1 ratio before seeding on tissue culture plastic.

**GlomSphere cell culture**. Spheroids were formed using a modified version of N3D bioscience standard protocol for magnetic spheroid bioprinting (Nano3D Biosciences Inc)(Greiner 655840). A T75 flask of cells was incubated overnight with 100 μl nanoshuttle-PL (Nano3D Biosciences Inc)(Greiner 657841), added to 10 ml of fresh cell culture medium. Cells were then washed with 5 ml sterile phosphate-buffered saline (PBS) and with trypsin-EDTA (Lonza). Cells were then pelleted via 1500 rpm centrifugation (5 mins) and counted with a Luna cell counter (Logos Biosystems). For monoculture experiments, 10,000 cells were pipetted into each well of an ultra-low attachment plate (Greiner) and a 96-magnet MagDrive (Nano3D Biosciences) was place underneath. Cells were left to form overnight at 33 °C.

For co-culture/GlomSphere spheroids, a spheroid of 5000 GEnCs was generated as above and allowed to aggregate for 1 h at 33 °C. The MagDrive was then removed and 5000 podocytes were pipetted into each well before the MagDrive was replaced. This forces the newly added podocytes to form a peripheral coating around the GEnC core. The spheroids were then thermoswitched to 37 °C and the podocyte layer migrates (Supplementary Fig. 1). After 10 days of differentiation, treatment or fixation was performed. Magnetic beads stay attached to the outer membrane of cells for the length of the experiment and are inert and non-fluorescent.

**Treatments with fibrotic and anti-fibrotic agents**. Under treatment conditions, GlomSphere cell culture media was supplemented with 10 ng/ml TGF-β1 (R&D

systems) and 10 µM Adriamycin (Sigma D1515) for 72 h. When stated, 10 µM Nintedanib was added at the same time as the fibrotic agent(s).

**Whole-mount fixation and immunofluorescent staining**. GlomSpheres were magnetically transferred to Eppendorf tubes and fixed in 4% Paraformaldehyde (Sigma) containing 1% Triton-x100 (Sigma) (20 mins, 20 °C). To block, spheroids were incubated with 5% bovine-serum-albumin (BSA) (Sigma) containing 0.1% Triton-x100 (overnight, 4 °C). Spheroids were then incubated with primary antibody, diluted in 5% BSA containing 0.1% Triton-x100 (48 h, 4 °C). Spheroids were then washed in PBS containing 1% Triton-x100 (3 × 30 min, room temp). Spheroids were then incubated with secondary fluorescent antibodies diluted 1:400 in 5% donkey serum (Sigma) containing 0.1% Triton-x100 (24 h, 4 °C). Spheroids were then washed in PBS containing 1% Triton-x100 (3 × 30 min, 20 °C).

**Sectioning and 2D immunofluorescent staining**. Sections were cut as previously described[34]. Sections or fixed 2D-cultured cells on glass coverslips were washed in PBS (2 × 5 mins, 20 °C). To block, samples were incubated in 5% BSA containing 0.1% Triton-x100 (45 min, 20 °C), then incubated with primary antibody, diluted in 5% BSA containing 0.1% Triton-x100 (1 h, 20 °C). Primary antibodies were then washed with PBS (2 × 5 mins, 20 °C) before being incubated with fluorescent secondary antibodies diluted in 1:400 in 5% donkey serum (1 h, 20 °C). Samples were then washed with PBS (2 × 5 mins, 20 °C) and mounted to microscope slides with 5 µl Mowiol (Sigma).

**Mid/high throughput screen**. Spheroids were formed in 96-well plates, using either GFP CI podocytes (WT) or GFP CI podocin mutant podocytes (PM). Compounds were added 24 h after culture and re-dosed daily. Images were taken daily and those shown and quantified are from day 5. Imaging was performed using an In-cell analyser (GE lifesciences).

**Antibodies**. Collagen IV pan (Abcam ab6586) (1:200), Col IVα1 and α3 (Chondrex 7070 and 7076, respectively) (1:100), Laminin α5 (Abcam ab77175) (1:200), fibronectin (Abcam ab2413) (1:50), podocin (Abcam ab50339) (0.2 µg/ml), nephrin (Origene BP5030)(1:50), PECAM-1 (Cell Signalling #3528)(1:800), podocalyxin (ThermoFisher 39-3800)(1:2000).

**Western blotting**. Two 96-well plates of spheroids were pooled (192 spheroids). Spheroids were disintegrated by mechanical force using a syringe and RIPA buffer (ThermoFisher) Standard western blots were then performed as previously described[34].

**Adhesion assay**. Cells are suspended in media at a concentration of $3 \times 10^5$/ml and incubated upright in a tube (10 min, 37 °C). Cells are added to the wells of a 96-well plate (50 µl/well) along with 50 µl of PBS and any inhibitor being tested. Leave cells to adhere (duration is cell dependent). Control wells (for 100% attachment) are then fixed in 4% PFA (20 mins, room temp). Wash all cells with (PBS 2× room temp) and fix remaining wells in 4% PFA (20 min, room temp). Wells are washed with 100 µl $H_2O$ (×3) before being stained with 0.1% crystal violet in 2% ethanol (60 mins, room temp). Cells are then washed with H2O (x3)

**QPCR**. Two 96-well plates of spheroids were pooled (192 spheroids). Spheroids were disintegrated by mechanical force using a syringe and RNA isolated with a RNeasy kit (Qiagen). cDNA was synthesized using a high capacity RNA to cDNA kit (Thermofisher).

Podocin forward: GTCCTCGCCTCCCTGATCTT
Podocin reverse: AGGAAGCAGATGTCCCAGTC
Nephrin forward: CCGGGAGACGCCTTAAACTT
Nephrin reverse: AGCCTTTGAATGGGGCTCTC
GAPDH forward: GAAGGTCGGAGTCAACGGATT
GAPDH reverse: CGCTCCTGGAAGATGGTGAT

**Imaging and analysis**. Confocal light microscopy was performed using SP8 spinning disc confocal microscope attachment (Leica), attached to a DMI 6000 inverted microscope (Leica). Widefield microscopy was performed on a Leica DM IRB linked to a Zeiss Axiocam ERc 5S. Lightsheet microscopy was performed using a Zeiss Z.1 lightsheet microscope.

All image analyses and quantifications were done using ImageJ software. For quantifications of IF images, a region of interest (ROI) was drawn around the measured structure (i.e. the perimeter of a spheroid) and mean fluorescence intensity in this area was calculated. In this way, pixels within the image that are absent of fluorescent signal are included in calculation, as well as the varying intensity of pixels positive for signal. In our experience this provides a more representative quantification than subjectively setting a threshold for the image and measuring total pixels present for a signal.

For podocyte coverage percentage, images were thresholded in imageJ and the same ROI was used to measure the percentage of spheroid area positive for

podocyte signal (and therefore a measure of relative podocyte adhesion to GlomSphere surface).

**Statistics and reproducibility**. Statistical analysis was performed using GraphPad Prism 7 software. Statistical tests used and sample sizes for each experiment are detailed in their corresponding figure legends. Unless stated otherwise, repeats are combined biological and technical repeats (that is, pooled and analysed spheroids from several experiments, each containing several spheroids).

**Patient plasma samples**. Patient plasma samples used are discarded plasma from patients undergoing plasmapheresis. They were obtained from patients, with ethical permission (National IRAS research ethics approval 09/H0106/80), from centres within the UK. Samples are immediately aliquoted and stored at −80 °C with freeze–thaw cycles avoided. Samples were taken from patients undergoing their first session of plasmapheresis for relapse or remission of FSGS[14]. Plasma is diluted in basal (non-supplemented) EBM-2 media at 15% concentration.

## Data availability
All raw data used to make graphs has been compiled and supplied as a Microsoft Excel spreadsheet (Supplementary Data 1). All unedited and uncropped western blot images (for Figs. 2 and 4) have been provided in Supplementary Fig. 7.

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

## Acknowledgements

This work was part funded by a BBSRC CASE studentship jointly with UCB pharmaceuticals (Slough). Additional funding was provided by Wellcome Trust ISSF, Medical Research Council (MR/R013942/1), the Elizabeth Blackwell Institute (TRACK) and the Nephrotic Syndrome Trust'. M.C. is funded by the Royal Thai Government Scholarship program. Confocal (BBSRC Alert 13 capital grant BB/L014181/1), Light sheet (Wellcome Trust), TEM and SEM (BBSRC 17ALERT) imaging was performed at the Wolfson Bioimaging Facility. TEM and SEM sample preparation and imaging was performed by Dr. Chris Neal and Mrs Judith Mantell.

## Author contributions

J.T. conducted and designed experiments, analysed data, created figures and wrote the manuscript. M.C. and V.K. provided additional data and experimental expertise. T.J. and UCB BioPharma provided funding and some materials. S.C. provided funding an expertise. G.I.W. and M.A.S. provided expertise, materials/equipment and funding and jointly designed experiments.

## Competing interests

T.J. and UCB Pharmaceuticals are using a variant of this model for drug screening. No other competing interests.
