## [Transparent Peer Review File · Communications Biology]

Reviewers' comments:

Reviewer #1 (Remarks to the Author):

Tuffin, et al. demonstrated an effective method for culturing conditionally immortalized human podocytes and GEnCs in 3D by forming spheroid cultures. The cells in these GlomSpheres segregate in an organized manner, they establish a GBM-like extracellular matrix with mature collagen IV at the podocyte/GEnC interface. The podocytes and GEnC were shown to be better differentiated and more mature than 2D cultures and have potential to be scaled for pharmaceutical screening. Authors show beautiful images that support the formation of FPs and higher levels of SD proteins. Overall, the manuscript is well written, the approach is adequate and findings support most of the author's conclusions. I have however some concerns:

1. As stated in the text the glomerulus is a complex filter, and although they investigate the interactions of cells within the filtration barrier; podocytes and GEnC, this method is limited as it's not a functional assay for measuring barrier function. Also, mechanistic studies cannot be performed given the combined cells in the assay. Authors must discuss these limitations.

2. Crosstalk occurs between podocytes and GEnC in the glomerular filtration barrier, as well as in a co-cultures (2D or transwell). A pathologic dialog indeed occurs in a disease setting, and this has been reported in FSGS and DKD, and surely contribute to ECM dysregulation. A discussion on intercellular crosstalk should be included.

3. Figure 5 compares monocultures in spheres, and co-cultured in 2D, these however are not described in methods or in the text. Please provide a description for these cultures.

4. In Figure 6, the authors note the podocyte changes that occur by loss of GFP detection in the combined treatments; TGFb and Adriamycin. Did they observe intensity changes in PECAM-1 in GEnC? This should be measured for combination and for TGFb only treatment.

5. Loss of podocyte GFP by TGFb and Adriamycin or NS plasma treatments is shown as a key measure for their assay. However, it is unclear how this is happening? Are cells detaching and found in the plates? Are podocytes losing GFP though changes/decreased in cytoskeleton? Are they undergoing cell death? Perhaps Tunel and WT1 co-staining would inform if cell death is taking place in podocytes. FACS would provide absolute number of cells as well as their viability after treatment.

6. The NS plasma treatment studies are informative, and clinically important. However, the effect of normal plasma was not examined, and this is an important control. Please provide the percent of plasma was used in the cultures in the Methods section and also the patient(s) clinical information.

Minor concerns

1. I could not find the supplemental Video 1.
2. Ref #32 is not cited in the text

Reviewer #2 (Remarks to the Author):

Tuffin et al. reported the generation of 3D spheres composed of a human immortalized podocyte cell line and a human immortalized glomerular endothelial cell (GEnC) line. The resulting GlomSpheres exhibited increased expression levels of podocyte and endothelial cell markers compared with 2D

cultured cells, and responded to Tgf-beta1 and adriamycin with increased ECM production and podocyte loss. Podocin mutant spheres also exhibited podocyte loss, which was restored to varying degrees by certain chemicals.

While I appreciate the increased expression levels of podocyte and endothelial cell markers in the 3D setting, the levels were still lower than those in vivo (or not compared) and the histologically distinct glomerular basement membranes and slit diaphragms were not formed. The quality of the whole-mount staining was sometimes insufficient to support the authors' claims. Consequently, the main readouts using GlomSpheres were whole-mount staining of collagen or GFP intensity (i.e. numbers) of podocytes, and the latter alone was applicable for medium-throughput screening, thereby limiting the usefulness of the protocol. Although a podocin mutant cell line showed fewer podocytes than the wild-type cell line, genetic background effects and clonal variations should also be considered.

Major comments

1. The statement "A scalable, flexible, 3D model" in the title is not supported by the results. Delete or tone down the claim.
2. Indicate clearly in the Abstract that immortalized cell lines were used. Otherwise, readers may misunderstand and think that primary human cells were used.
3. Last sentence in the first results section: The meaning of "induce this organisation manually for reproducibility via layering spheroids" is unclear. Indicate clearly which method was utilized for all main figures: aggregation of endothelial cells followed by podocytes (Supplementary Figure 1) or aggregation of mixed endothelial cells and podocytes (Supplementary Figure 2). The main text indicates the latter, while the Methods section appears to indicate the former.
4. Figure 3: Present the TEM section for the glomerular basement membrane between podocytes and endothelial cells. This is essential to claim that Col4a5 is deposited in the basement membrane at the podocyte/GEEnC interface. In contrast, Figure 3f shows non-banded collagen in a lumen-like space. This discrepancy should be addressed.
5. Figure 4: Compare the expression levels of Podocalyxin and Pecam-1 between glomeruli in vivo and GlomSpheres, and state the limitation of GlomSpheres that their Podocalyxin level is low. Also present section staining to demonstrate clear Podocalyxin expression in the podocytes and neighboring endothelial cells positive for Pecam-1.
6. Figure 5: Whole-mount staining of Nephhrin and Podocin is shown, but the signal specificity and relative expression levels to those in vivo remain unclear. Present western blots as in Figure 4. Inclusion of glomeruli in vivo in the qPCR analyses (panels i, j) would also be helpful. Regarding Supplementary Figure 4, higher-power magnification images of the stained sections would be informative to show the distributions of Nephhrin on the basal side of podocytes facing the Col4+ basement membrane and Pecam-1+ endothelial cells.
7. Describe how the serum samples were obtained (including details of ethical approval), how the serum was processed (with or without complement inactivation), and how much dilution was employed. Also describe the effects of serum samples from healthy individuals as controls.
8. The immortalized Podocin mutant cell line and the wild-type cell line have different genetic backgrounds, but one cell line per group was compared. Ideally, multiple clones, as well as genome-edited clones, should be examined. At least, genetic background effects and clonal variability should be discussed.
9. Although Podocin mutations led to impaired slit diaphragm formation in podocytes in vivo, Figure 8 showed that reduced numbers of podocytes remained attached to the spheres. Thus, the measurements in GlomSpheres appeared to reflect the adhesive capacity of podocytes, similar to the case for 2D culture, rather than the quality of podocytes in vivo (such as slit diaphragm formation). This may explain the similar tendencies in the responses to chemicals between the 3D and 2D settings, although the dynamic ranges may be different. These features limit the usefulness of the protocol.

Minor comments

1. Because magnetic spheroids were eluted in the Abstract, describe the principle of how the magnetic

spheroid method was utilized in the Results section.

2. Main text and Figure 1 legend: Explain why podocytes and endothelial cells are green and red, respectively. It is difficult to understand that the color tags were overexpressed until the Methods section is consulted.

3. Figure 2: Present western blots for Lama5a.

4. Main text related to Figure 4: pe-1 is a typo of Pecam-1.

5. Figure 8: Describe the differences between GFP mean fluorescence (panel d) and integrated density (panel e).

Reviewer #3 (Remarks to the Author):

The authors describe a new method for generating so-called Glomspheres, a 3D tissue generation by a combination of GeNCs with podocytes. The ability to generate 3D structures for is promising for disease modelling and screening therapeutic compounds. Especially the detection of mature COL4 in these tissues is promising.

While the manuscript is well written, I do not find clear structure in the text and I would advise the authors to combine several figures (for example generation and characterisation of glomspheres / 2D vs 3D culture etc). That would make the story more clear. The end of the paper looking at disease models and pharmaceutical screening has a stronger line and goal.

Furthermore, the authors base many of their quantification and conclusions on the intensity of fluorescent signals, but I doubt whether this is a good approach.

Furthermore I have other additional concerns:

Major:

Could the authors explain more about the method to generate the Glomspheres? It would be informative to include a schematic of the protocol and use of the magnetic beads.

From Figure 1 it seems that the podocytes mostly surround the top part of the spheroid and that the border on the top left side of Figure 1a-d was located at the bottom of the 96 well plate. Is that right?

Figure 1e shows more a mix of cells and not so much a vascular center with podocytes surrounding – also COL IV is not so well visible here. I do not see the staining extracellularly in both cell types.

Can the authors make a statement in the text about the influence of the beads on the cells? Do they stay attached for prolonged time (only a few days, or for the entire duration of the experiment)? Would V-shaped plates have been an option too? Do the beads show autofluorescence that could interfere with the analysis?

What do the authors mean by 'induce this organisation manually' (page 5)? What did the authors do?

Supplemental figure 2: have the authors also looked at collagen in the smaller spheroids? Would they continue to grow?

Laminin is also an important component of the GBM. Have the authors also looked at this?

Why have the authors stained for fibronectin? Which cell produces fibronectin? Would it also be produced without the interaction between GeNCs and podocytes?

Figure 2: I am wondering if quantification of the intensity is a fair read out for amount of collagen IV. Why did the authors chose for intensity? Intensity can be highly variable when looking at whole mount structures. Would a quantification based on presence or absence be better? Are we looking at slides or whole mount? What are the red dots in the pictures? The intensity will be highly influenced them.

Have the 'podocytes only' been cultured in EMV2 medium? Does this support those cells enough?

Figure 2m,n: The western blots are more informative, but I do not think the bars for gapdh are comparable between the different co-cultures. Especially COL4a1 is very faint in the co-culture, and we could miss that in the podocyte only blot.

Figure 3: the SEM images look impressive and give a good overview of the structure itself. It is described in the manuscript text that the deposition of collagen IV is visible in e and f. The authors cannot be sure that this indeed is COL IV.

The authors compare 2D and glomspheres. How have the cells been cultured in 2D? This is neither described in results, nor in methods.

Figure 4: the presence of podocalyxin and pecam is present based on a and b, but the staining in c is not very specific. Have the authors compared this with a positive control (human glomerular structure?)

Why is pecam1 in Figure 4d and 5a,e so different?

Figure 8a-e: could podocytes be dying in the mutant? What do the authors mean by: attach more strongly? To each other? To the endothelial cells?
Have the authors looked at podocin expression and restoration? It would be helpful to show podocin rescue in these glomspheres (as mentioned as manuscript in submission).
Can the authors include pictures of multiple treated glomspheres treated?

Minor comments:

In several figures it would be easier for interpretation if a brief description is included in the figure (for pictures and plots). For example Figure 2 a-i: include details of the conditions on the left side of the IF pictures describing the 3 different types of spheroids and Figure 6i-j for plots. This helps so that the reader does not need to read and search in the (long) legends for the different conditions. Just + or - compound, what is measured and names for the colours etc. would be helpful.

Figure 2j: p-value is missing (says INSERT HERE)

Supplemental video 1 was not shared with me.

Figure 6: first time use of abbreviation MIP

Reviewer #1 (Remarks to the Author):

Tuffin, et al. demonstrated an effective method for culturing conditionally immortalized human podocytes and GEnCs in 3D by forming spheroid cultures. The cells in these GlomSpheres segregate in an organized manner, they establish a GBM-like extracellular matrix with mature collagen IV at the podocyte/GEnC interface. The podocytes and GEnC were shown to be better differentiated and more mature than 2D cultures and have potential to be scaled for pharmaceutical screening. Authors show beautiful images that support the formation of FPs and higher levels of SD proteins. Overall, the manuscript is well written, the approach is adequate and findings support most of the author's conclusions. I have however some concerns:

1. As stated in the text the glomerulus is a complex filter, and although they investigate the interactions of cells within the filtration barrier; podocytes and GEnC, this method is limited as it's not a functional assay for measuring barrier function. Also, mechanistic studies cannot be performed given the combined cells in the assay. Authors must discuss these limitations.

Response: The GlomSphere model seeks to improve on the industry standard 2D assays, which are also not capable of measuring barrier function. We do however agree that this is a limitation of the model and have added the following to the discussion:

“The GlomSphere model is also currently not a model of glomerular function (i.e., does not model filtration efficiency). We are currently exploring possibilities to modify the model to this end, although scalability is likely to be compromised in this case.”

2. Crosstalk occurs between podocytes and GEnC in the glomerular filtration barrier, as well as in a co-cultures (2D or transwell). A pathologic dialog indeed occurs in a disease setting, and this has been reported in FSGS and DKD, and surely contribute to ECM dysregulation. A discussion on intercellular crosstalk should be included.

Response: We agree and this was removed in original submission to be succinct. Have now added the following to introduction:

“For this reason, reductionist 2D cell models of disease are employed in pre-clinical testing, whereby they are used to narrow down libraries of >5000 compounds (15). Many of these models are monoculture, and thus fail to replicate the intricate crosstalk environment of the glomerular filtration barrier that is understood to be important in glomerular disease progression (1, 2).”

3. Figure 5 compares monocultures in spheres, and co-cultured in 2D, these however are not described in methods or in the text. Please provide a description for these cultures.

Response: Have added the following section to methods, and have updated supplementary figure 1 for clarity:

“2D co-culture” conditions in experiments are achieved by mixing podocytes and GEnCs at a 1:1 ratio before seeding on tissue culture plastic.”

4. In Figure 6, the authors note the podocyte changes that occur by loss of GFP detection in the combined treatments; TGFb and Adriamycin. Did they observe intensity changes in PECAM-1 in GEnC? This should be measured for combination and for TGFb only treatment.

Response: We agree that changes in endothelial vasculature are important. Previous experiments showed minimal differences to PECAM-1 intensity, which we believe was hard to measure due to its location at the centre of the GlomSphere. There do however appear to be changes in vascularisation in the presence of various disease plasma conditions, which are the focus of ongoing work. We hope to share these when complete in an additional publication.

5. Loss of podocyte GFP by TGF β and Adriamycin or NS plasma treatments is shown as a key measure for their assay. However, it is unclear how this is happening? Are cells detaching and found in the plates? Are podocytes losing GFP through changes/decreased in cytoskeleton? Are they undergoing cell death? Perhaps Tunel and WT1 co-staining would inform if cell death is taking place in podocytes. FACS would provide absolute number of cells as well as their viability after treatment.

Response: Agree that this was unclear in the text. Have now added an additional supplementary figure (Supplementary Figure 5), which shows cells being lost into surrounding media (before media is changed like in other images), as well as a viability assay of these detached cells, which shows them to be around 36.8% viable).

Figure legend reads: "Clarification of podocyte loss following injurious stimulation. The untreated spheroid (left) is shown to have intact podocyte coverage (green), with no cells lost into the surrounding medium. The TGF β 1 + Adriamycin treated spheroid is shown to have begun losing podocytes into the surrounding medium, which are removed upon changing media. A cell count of cells lost in this way (with the addition of trypan blue) indicates that these cells were ~36.8% viable at the time of counting."

6. The NS plasma treatment studies are informative, and clinically important. However, the effect of normal plasma was not examined, and this is an important control. Please provide the percent of plasma used in the cultures in the Methods section and also the patient(s) clinical information.

Response: We agree that the response of 'normal' plasma is important. Unfortunately healthy patients do not undergo plasmapheresis, as with our relapse and remission condition. Healthy plasma from blood draw is routinely collected in lithium-heparin or EDTA tubes, which we have shown in separate signalling experiments to obliterate the effects of plasma on podocytes. We do however believe that because "relapse" and "remission" plasma is taken from the same patient at different dates, they serve as an internal control and are preferable given the unavailability of a "normal" patient. Have included the following in the text.

"Whilst it would be optimal to compare relapse and remission patients to a "normal" control patient, healthy patients do not undergo plasmapheresis like FSGS patients. For this reason, we judged the internal control offered by examining the same patient at different dates (in relapse and remission) to be the fairest control."

Minor concerns

1. I could not find the supplemental Video 1.

Response: Will ensure this is received.

2. Ref #32 is not cited in the text

Response: Removed

Reviewer #2 (Remarks to the Author):

Tuffin et al. reported the generation of 3D spheres composed of a human immortalized podocyte cell line and a human immortalized glomerular endothelial cell (GEnC) line. The resulting GlomSpheres exhibited increased expression levels of podocyte and endothelial cell markers compared with 2D cultured cells, and responded to Tgf-beta1 and adriamycin with increased ECM production and podocyte loss. Podocin mutant spheres also exhibited podocyte loss, which was restored to varying degrees by certain chemicals.

While I appreciate the increased expression levels of podocyte and endothelial cell markers in the 3D setting, the levels were still lower than those in vivo (or not compared) and the histologically distinct glomerular basement membranes and slit diaphragms were not formed. The quality of the whole-mount staining was sometimes insufficient to support the authors' claims. Consequently, the main readouts using GlomSpheres were whole-mount staining of collagen or GFP intensity (i.e. numbers) of podocytes, and the latter alone was applicable for medium-throughput screening, thereby limiting the usefulness of the protocol. Although a podocin mutant cell line showed fewer podocytes than the wild-type cell line, genetic background effects and clonal variations should also be considered.

Major comments

1. The statement "A scalable, flexible, 3D model" in the title is not supported by the results. Delete or tone down the claim.

Response: We feel the results do support the title but will leave the final decision to the Editors/Reviewers, and are happy to accept the advice. To support our view, whilst we do not scale our experiments to their maximum potential in figure 8, we show evidence of the model's scalability by demonstrating its compatibility with the in-cell analyser, which is used for medium and high throughput assays. With regards to claims of flexibility, we demonstrate the model's responsiveness to several insults, including TGF- β 1, Adriamycin and relapse and remission FSGS patient plasma. We also demonstrate incorporation of mutant patient cell lines in to GlomSpheres, illustrating its potential to model genetic kidney disease variants.

2. Indicate clearly in the Abstract that immortalized cell lines were used. Otherwise, readers may misunderstand and think that primary human cells were used.

Response: We agree and the abstract has now been changed to reflect this.

Abstract section now reads:

"Here we report a rapidly generated and highly reproducible 3D co-culture spheroid model (GlomSpheres), better demonstrating the specialised physical and molecular structure of a glomerulus. Co-cultured using a magnetic spheroid formation approach, conditionally immortalised (CI) human podocytes and glomerular endothelial cells (GEnCs) deposited mature, organized isoforms of collagen IV and Laminin."

3. Last sentence in the first results section: The meaning of “induce this organisation manually for reproducibility via layering spheroids” is unclear. Indicate clearly which method was utilized for all main figures: aggregation of endothelial cells followed by podocytes (Supplementary Figure 1) or aggregation of mixed endothelial cells and podocytes (Supplementary Figure 2). The main text indicates the latter, while the Methods section appears to indicate the former.

Response: We agree that this was misleading. “Manual organisation” refers to the sequential layering demonstrated in supplementary figure 1, but supplementary figure 2 is referenced. This has been changed to supplementary figure 1. Figure 1 itself has been changed for clarity also.

4. Figure 3: Present the TEM section for the glomerular basement membrane between podocytes and endothelial cells. This is essential to claim that Col4a5 is deposited in the basement membrane at the podocyte/GEnC interface. In contrast, Figure 3f shows non-banded collagen in a lumen-like space. This discrepancy should be addressed.

Response: We agree that this discrepancy needed addressing and have undertaken additional work to this end. Figure 3f has now been added, which shows fibrillar collagen between a podocyte and what appears to be a GEnC. This is in line with IF which shows collagen IV at the podocyte/GEnC interface.

5. Figure 4: Compare the expression levels of Podocalyxin and Pecam-1 between glomeruli in vivo and GlomSpheres, and state the limitation of GlomSpheres that their Podocalyxin level is low. Also present section staining to demonstrate clear Podocalyxin expression in the podocytes and neighboring endothelial cells positive for Pecam-1.

Response: Whilst we agree that section staining would further support the whole mount stain, we have presented co-staining of podocyte proteins (nephrin and podocin) with PECAM-1 in Figure 5, which we believe demonstrates the same point – that of the distinct anatomical localisation between podocytes and endothelial cells.

We agree that GlomSphere podocalyxin expression is low in comparison to glomerular lysate, the improvement over the 2D lysate (which is the current gold standard in vitro approach) is however encouraging. An addition to figure 5's explanation now reads:

“Whilst podocalyxin appears to be absent in the 2D lysate, it is significantly upregulated in GlomSpheres (~6 fold \pm 0.3) (Tukey's multiple comparison **** P =<0.0001) (Fig 4a). Compared to the glomerular control, podocalyxin is lower in GlomSpheres (~10 fold increase in Glomerulus VS ~6 fold increase in GlomSphere) (Tukey's multiple comparison **** P =<0.0001).”

6. Figure 5: Whole-mount staining of Nephrin and Podocin is shown, but the signal specificity and relative expression levels to those in vivo remain unclear. Present western blots as in Figure 4. Inclusion of glomeruli in vivo in the qPCR analyses (panels i, j) would also be helpful. Regarding Supplementary Figure 4, higher-power magnification images of the stained sections would be informative to show the distributions of Nephrin on the basal side of podocytes facing the Col4+ basement membrane and Pecam-1+ endothelial cells.

Response: We agree that higher power images to show nephrin and podocin localisation would be ideal, but these antibodies are known to have low sensitivity, so are not discriminatory enough in IF to distinguish basal cell expression, and are not clean enough for western blot. For this reason, we used QPCR to demonstrate and quantify expression levels instead. A comparison between GlomSpheres and human kidney cortex expression of podocin and nephrin has now been made (supplementary figure 6). New text now reads:

A comparison of GlomSphere and human kidney cortex expression of podocin and nephrin was also made as a positive control for primers (Supplementary figure 6). Despite variability between cortex samples (likely due to differing glomeruli counts), expression levels were stronger than GlomSpheres (more so for nephrin than podocin, which was closer to in vivo expression).

7. Describe how the serum samples were obtained (including details of ethical approval), how the serum was processed (with or without complement inactivation), and how much dilution was employed. Also describe the effects of serum samples from healthy individuals as controls.

Response: We agree that failing to include this was an oversight. The following has now been added to the methods section.

“Patient plasma samples used are discarded plasma from patients undergoing plasmapheresis. They were obtained from patients, with ethical permission (National IRAS research ethics approval 09/H0106/80), from centres within the UK. Samples are immediately aliquoted and stored at -80°C with freeze–thaw cycles avoided. Samples were taken from patients undergoing their first session of plasmapheresis for relapse or a subsequent session once achieving remission of FSGS (14). Plasma is diluted in basal (non-supplemented) EBM-2 media at 15% concentration.”

8. The immortalized Podocin mutant cell line and the wild-type cell line have different genetic backgrounds, but one cell line per group was compared. Ideally, multiple clones, as well as genome-edited clones, should be examined. At least, genetic background effects and clonal variability should be discussed.

Response: We agree that this should have been clearer. The following text has been added to the methods section:

“Each individual cell line is from a different single patient, and as such, are from different genetic backgrounds. Several clones of each are taken and used interchangeably in our lab. No difference between clones is observed.”

9. Although Podocin mutations led to impaired slit diaphragm formation in podocytes in vivo, Figure 8 showed that reduced numbers of podocytes remained attached to the spheres. Thus, the measurements in GlomSpheres appeared to reflect the adhesive capacity of podocytes, similar to the case for 2D culture, rather than the quality of podocytes in vivo (such as slit diaphragm formation). This may explain the similar tendencies in the responses to chemicals between the 3D and 2D settings, although the dynamic ranges may be different. These features limit the usefulness of the protocol.

Response: We agree that the “adhesive capacity” of podocytes is not a direct homologue to slit diaphragm formation. We have clarified in the text as follows:

“A podocyte cell-line generated from a patient with a pathogenic R138Q mutation of podocin (14) was used for these experiments to study the ability of various compounds to rescue podocin functionality. Our 2D experiments on this cell line shows decreased adhesion compared to wild-type podocytes (unpublished data). Podocyte loss from the spheroid was quantified as a surrogate for podocyte adhesiveness (which is a well established readout of podocyte pathology in in vitro assays) and was compared in wildtype (WT), podocin mutant (PM) and PM cells treated with various compounds.”

This reviewer is correct in pointing out that the response to chemicals is similar in 2D vs 3D, which is exactly what we were attempting to achieve – the current gold standard is 2D, which is laborious, and therefore the advantage demonstrated is to achieve much higher throughput, as well as enhance the dynamic range of the experiment.

Minor comments

1. Because magnetic spheroids were eluted in the Abstract, describe the principle of how the magnetic spheroid method was utilized in the Results section.

Response: Agree. Use of magnetic spheroid technique is made clearer throughout.

2. Main text and Figure 1 legend: Explain why podocytes and endothelial cells are green and red, respectively. It is difficult to understand that the color tags were overexpressed until the Methods section is consulted.

Response: Fig 1 legend now reads:

“Immunofluorescence imaging of GlomSpheres in 2D paraffin sections (a-d) and 3D whole-mount (e). (a) Central core of GEnCs (overexpressing M-cherry to aid localisation) (b) Outer layer of podocytes (overexpressing GFP to aid localisation) “

3. Figure 2: Present western blots for Lama5a.

Response: Whilst we agree that this is missing from this figure, the Lama5a antibody used for this experiment is not validated for western blotting. Several attempts were made to use this antibody for western blot, unsuccessfully.

4. Main text related to Figure 4: pe-1 is a typo of Pecam-1.

Response: Have corrected this.

5. Figure 8: Describe the differences between GFP mean fluorescence (panel d) and integrated density (panel e).

Response: Text now reads “integrated density, which is the product of mean intensity and GlomSphere area (Fig 8e).” This has also been added to figure legends.

Reviewer #3 (Remarks to the Author):

The authors describe a new method for generating so-called Glomspheres, a 3D tissue generation by a combination of GEnCs with podocytes. The ability to generate 3D structures for is promising for disease modelling and screening therapeutic compounds. Especially the detection of mature COL4 in these tissues is promising.

While the manuscript is well written, I do not find clear structure in the text and I would advise the authors to combine several figures (for example generation and characterisation of glomspheres / 2D vs 3D culture etc). That would make the story more clear. The end of the paper looking at disease models and pharmaceutical screening has a stronger line and goal.

Furthermore, the authors base many of their quantification and conclusions on the intensity of fluorescent signals, but I doubt whether this is a good approach.

Furthermore I have other additional concerns:

Major:

Could the authors explain more about the method to generate the Glomspheres? It would be informative to include a schematic of the protocol and use of the magnetic beads.

Response: Agree that this needed clarification. "Manual organisation" is explained better throughout with references to supplementary figure 1. The step by step process of first forming a GEnC core, which is then wrapped by podocytes is illustrated clearly here, with a schematic for additional clarity. The figure legend now reads:

"Formation and reorganisation of glomerular spheroids. (A) Sequence of spheroid formation. A core of GEnCs (red) is first formed from 5,000 cells. After an hour to stabilise, a peripheral coating of podocytes (green) is then wrapped around the GEnC core, forming a spheroid with a distinct boundary between the two cell types. A schematic representation shows this process more clearly."

From Figure 1 it seems that the podocytes mostly surround the top part of the spheroid and that the border on the top left side of Figure 1a-d was located at the bottom of the 96 well plate. Is that right?

Response: Figure 1a-d are paraffin embedded sections of spheroids (2D slices of a 3D structure) and it is very difficult to determine initial positioning in the well. Supplementary figure 1b does however show that initially the "side" of the spheroid is coated, with the top being covered as podocytes migrate over the course of differentiation.

Figure 1e shows more a mix of cells and not so much a vascular center with podocytes surrounding – also COL IV is not so well visible here. I do not see the staining extracellularly in both cell types.

Response: Agree that vascular centre is less visible in this image. Microscopy used here was confocal and not lightsheet (as in figure 5) which has better Z-plane resolution. This confocal image does however show surface podocytes in greater detail (due to enhanced x-y plane resolution).

Can the authors make a statement in the text about the influence of the beads on the cells? Do they stay attached for prolonged time (only a few days, or for the entire duration of the experiment)?

Would V-shaped plates have been an option too? Do the beads show autofluorescence that could interfere with the analysis?

Response: The following is now added to the methods section:

“Magnetic beads stay attached to the outer membrane of cells for the length of the experiment and are inert and non-fluorescent.”

What do the authors mean by ‘induce this organisation manually’ (page 5)? What did the authors do?

Response: We agree that this was not clear and has been clarified in supplementary figure 1 (see response above)

Supplemental figure 2: have the authors also looked at collagen in the smaller spheroids? Would they continue to grow?

Response: Smaller spheroids continue to mature, but do not grow in size and we have not stained them. The spheroids of around 200µm are closer to in vivo glomerular size, which we believe is of more interest from a modelling perspective.

Laminin is also an important component of the GBM. Have the authors also looked at this? Why have the authors stained for fibronectin? Which cell produces fibronectin? Would it also be produced without the interaction between GeNCs and podocytes?

Response: See figure 2 c, f and i for mature laminin A5 staining. Both cell types produce fibronectin in vivo and in spheroid culture (1, 2)

Figure 2: I am wondering if quantification of the intensity is a fair read out for amount of collagen IV. Why did the authors chose for intensity? Intensity can be highly variable when looking at whole mount structures. Would a quantification based on presence or absence be better? Are we looking at slides or whole mount? What are the red dots in the pictures? The intensity will be highly influenced them.

Response: We thank the reviewer for pointing this out. Region of interest intensity was not clear in original submission and has been clarified. A region is drawn around the perimeter of a spheroid, meaning that all pixels within this are quantified. This allows for pixels absent and present for fluorescent signal to be weighed into calculation. The following has been added to the text:

“For quantifications of IF images, a region of interest was drawn around the measured structure (i.e the perimeter of a spheroid) and mean fluorescence intensity in this area was calculated. In this way, pixels within the image that are absent of fluorescent signal are included in calculation, as well as the varying intensity of pixels positive for signal. In our experience this provides a more representative quantification than subjectively setting a threshold for the image and measuring total pixels present for a signal.”

Have the ‘podocytes only’ been cultured in EMV2 medium? Does this support those cells enough?

Response: Yes podocyte EBM-2 media used for podocyte monoculture and experiments suggest this supports podocytes well, with no change in phenotype or viability. The same cannot be said for culturing GEnCs in RPMI-1640. We have now clarified that this was determined by preliminary experiments.

Figure 2m,n: The western blots are more informative, but I do not think the bars for gapdh are comparable between the different co-cultures. Especially COL4a1 is very faint in the co-culture, and we could miss that in the podocyte only blot.

Response: Have adjusted brightness/contrast of this image to make it easier to discern.

Figure 3: the SEM images look impressive and give a good overview of the structure itself. It is described in the manuscript text that the deposition of collagen IV is visible in e and f. The authors cannot be sure that this indeed is COL IV.

Response: We agree that we cannot be sure this is Col IV. Have since added an additional TEM image showing fibrillar collagen between what appears to be a podocyte and GEnC. This is further supportive of this being Col IV, but clearly not conclusive. The following text has been added to the TEM results section:

“A tri-layer structure is shown in Fig 3g, whereby fibrillar collagen (yellow arrows) is deposited in the extracellular space between a neighbouring podocyte and endothelial cell.”

The authors compare 2D and glomspheres. How have the cells been cultured in 2D? This is neither described in results, nor in methods.

Response: Agree and the methods now reads:

“2D co-culture” conditions in experiments are achieved by mixing podocytes and GEnCs at a 1:1 ratio before seeding on tissue culture plastic”

Figure 4: the presence of podocalyxin and pecam is present based on a and b, but the staining in c is not very specific. Have the authors compared this with a positive control (human glomerular structure?)

Response: This particular image was taken using widefield microscopy, whereas in Fig 5 we obtained much more specific staining with podocyte specific markers nephrin and podocin using confocal slice images. We felt the latter molecules would be a better demonstration of podocyte marker expression and distribution compared to podocalyxin. Therefore we agree that comparing podocalyxin in GlomSpheres vs human glomerulus would be interesting, but not substantially additive for the purposes here.

Why is pecam1 in Figure 4d and 5a,e so different?

Response: Figure 4 was taken using widefield microscopy and represents a single focal plane, figure 5 is a lightsheet image and has far greater resolution.

Figure 8a-e: could podocytes be dying in the mutant? What do the authors mean by: attach more strongly? To each other? To the endothelial cells?

Response: We agree that this was unclear in the text. Have now added an additional supplementary figure (Supplementary Figure 5), which shows cells being lost into surrounding media (before media is changed like in other images), as well as a viability assay of these detached cells, which shows them to be around 36.8% viable). We observe the same thing with podocin mutant cells.

Figure legend reads: "Clarification of podocyte loss following injurious stimulation. The untreated spheroid (left) is shown to have intact podocyte coverage (green), with no cells lost into the surrounding medium. The TGF β 1 + Adriamycin treated spheroid is shown to have begun losing podocytes into the surrounding medium, which are removed upon changing media. A cell count of cells lost in this way (with the addition of trypan blue) indicates that these cells were ~36.8% viable at the time of counting."

Have the authors looked at podocin expression and restoration? It would be helpful to show podocin rescue in these glomspheres (as mentioned as manuscript in submission). Can the authors include pictures of multiple treated glomspheres treated?

Response: A genetic restoration of podocyte is a key focus of much of our unpublished work and we agree that showing its restoration in GlomSpheres will be helpful. This work is currently being worked up for submission as a separate manuscript, wherein we have discovered a small molecule that rescues podocin missense mutation, localisation and function.

Minor comments:

In several figures it would be easier for interpretation if a brief description is included in the figure (for pictures and plots). For example Figure 2 a-i: include details of the conditions on the left side of the IF pictures describing the 3 different types of spheroids and Figure 6i-j for plots. This helps so that the reader does not need to read and search in the (long) legends for the different conditions.

Response: Agree and have modified figures 2, 6 and 7 for clarity.

Just + or – compound, what is measured and names for the colours etc. would be helpful.

Figure 2j: p-value is missing (says INSERT HERE)

Response: Have corrected this.

Supplemental video 1 was not shared with me.

Response: Will ensure this is shared for resubmission

Figure 6: first time use of abbreviation MIP

Response: Have changed this to "Max intensity projection".

1. Satchell SC, Tasman CH, Singh A, Ni L, Geelen J, von Ruhland CJ, et al. Conditionally immortalized human glomerular endothelial cells expressing fenestrations in response to VEGF. *Kidney Int.* 2006;69(9):1633-40.
2. Kliewe F, Kaling S, Löttsch H, Artelt N, Schindler M, Rogge H, Schröder S, Scharf C, Amann K, Daniel C, Lindenmeyer MT, Cohen CD, Endlich K, Endlich N. Fibronectin is up-regulated in podocytes

by mechanical stress. FASEB J. 2019 Dec;33(12):14450-14460. doi: 10.1096/fj.201900978RR. Epub 2019 Nov 1. PMID: 31675484.

Reviewers' comments:

Reviewer #1 (Remarks to the Author):

The authors have adequately addressed all of my comments. I would suggest to consider using log scales for Figures 5i and 5j.

Reviewer #2 (Remarks to the Author):

The authors have addressed most of my concerns.

Minor comments

The three panels in Supplementary figure 5 should be labeled (a, b, c) with the corresponding explanations in the legend. I also suggest the authors improve the quality of Panel c.

Reviewer #3 (Remarks to the Author):

The authors have answered the comments from all reviewers in writing and adjusted the text following the suggestions by the reviewers. However (almost) no additional experiments were performed. The added supplemental figures assist in the interpretation of the model and analysis. I do think some additional experiments would have been helpful.

I am still not sure whether fluorescent intensity is a good measure for a large part of the analysis. I would advise the authors to see if another quantification (such as volume measurement, or percentage covered) is possible as an alternative.

Reviewers' comments:

Reviewer #1 (Remarks to the Author):

The authors have adequately addressed all of my comments. I would suggest to consider using log scales for Figures 5i and 5j.

Author Response: We believe that the raw $2^{-\Delta\Delta Ct}$ values with a split Y-axis are more intuitive than log scaled values and have decided to keep the original figure.

Reviewer #2 (Remarks to the Author):

The authors have addressed most of my concerns.

Minor comments

The three panels in Supplementary figure 5 should be labeled (a, b, c) with the corresponding explanations in the legend. I also suggest the authors improve the quality of Panel c.

Author Response: We agree that supplementary figure 5 was not labelled adequately and this has now been modified to include a, b and c labels. The legend has also been updated to reflect this.

Reviewer #3 (Remarks to the Author):

The authors have answered the comments from all reviewers in writing and adjusted the text following the suggestions by the reviewers. However (almost) no additional experiments were performed. The added supplemental figures assist in the interpretation of the model and analysis. I do think some additional experiments would have been helpful.

I am still not sure whether fluorescent intensity is a good measure for a large part of the analysis. I would advise the authors to see if another quantification (such as volume measurement, or percentage covered) is possible as an alternative.

Author Response: We apologize for previously misunderstanding which parts of the analysis this referred to. We have interpreted this comment to be in reference to measures of podocyte attachment to GlomSphere surface. We agree that as this fluorescence is not from a fluorescent antibody signal (but from the endogenous GFP in the podocyte cells) that intensity is not as important.

We have now included "podocyte coverage %" panels to figure 6 (e, l), figure 7 (h) and figure 8 (e). respective legends and text have been updated to describe relevant data. The following has been added to the methods section:

"For podocyte coverage percentage, images were thresholded in imageJ and the same ROI was used to measure the percentage of spheroid area positive for podocyte signal (and therefore a measure of relative podocyte adhesion to GlomSphere surface)."

REVIEWERS' COMMENTS:

None remaining